Li *et al. Genome Biology*     (2021) 22:221

**METHOD**

# SCAPTURE: a deep learning-embedded pipeline that captures polyadenylation information from 3′ tag-based RNA-seq of single cells

Guo-Wei Li[1†], Fang Nan[1†], Guo-Hua Yuan[1], Chu-Xiao Liu[2], Xindong Liu[3], Ling-Ling Chen[2,4,5], Bin Tian[6] and Li Yang[1,4*]

* Correspondence: liyang@picb.ac.
cn
†Guo-Wei Li and Fang Nan
contributed equally to this work.
[1]CAS Key Laboratory of
Computational Biology, Shanghai
Institute of Nutrition and Health,
University of Chinese Academy of
Sciences, Chinese Academy of
Sciences, Shanghai 200031, China
[4]School of Life Science and
Technology, ShanghaiTech
University, Shanghai 201210, China
Full list of author information is
available at the end of the article

## Abstract

Single-cell RNA-seq (scRNA-seq) profiles gene expression with high resolution. Here, we develop a stepwise computational method-called SCAPTURE to identify, evaluate, and quantify cleavage and polyadenylation sites (PASs) from 3′ tag-based scRNA-seq. SCAPTURE detects PASs de novo in single cells with high sensitivity and accuracy, enabling detection of previously unannotated PASs. Quantified alternative PAS transcripts refine cell identity analysis beyond gene expression, enriching information extracted from scRNA-seq data. Using SCAPTURE, we show changes of PAS usage in PBMCs from infected versus healthy individuals at single-cell resolution.

**Keywords:** scRNA-seq, PAS, APA, Deep learning, Peak calling, Transcript quantification

## Introduction

The advent of single-cell RNA-seq (scRNA-seq) has enabled gene expression analysis with an unprecedented resolution [1, 2]. Based mainly on differential gene expression (DGE) [3], scRNA-seq reveals heterogeneity within a bulk of cells [4], complex tissues [5–8], or even the whole animal [9], resulting in the identification of distinct cell identities and lineage trajectories, especially in developing or differentiating systems [10, 11]. By taking advantage of machine-based cell isolation, hundreds of thousands of cells can now be individually processed for RNA enrichment and deep sequencing analysis [12, 13]. Recently, scRNA-seq technologies that use oligo(dT) priming for cDNA generation and subsequent short-read sequencing from their 3′-ends (herein called 3′ tag-based scRNA-seq) [14], such as inDrops [15], Drop-seq [12], Seq-Well [16], and 10x Chromium [13], have been broadly adopted. In contrast to full-length scRNA-seq (such as Smart-seq2) and canonical bulk cell RNA-seq (such as Illumina TruSeq), these 3′ tag-based scRNA-seq data characteristically have an enrichment of reads at the 3′

ends of genes. For example, a comparison of transcriptome profiling in human PBMCs with TruSeq, Smart-seq2, and 10x Chromium [17] showed that a bias of mapped reads to annotated cleavage and polyadenylation sites (PASs) in the 10x Chromium data for both protein-coding (such as *GAPDH*, Fig. 1a, b) and noncoding (such as *NORAD* and *GAS5*, Additional file 1: Fig. S1A) genes. By contrast, TruSeq and Smart-seq2 data profile gene expression with reads covering the whole gene body (Fig. 1a, b; Additional file 1: Fig. S1B). As such, 3′ tag-based scRNA-seq datasets can be mined for PAS identification and for expression of transcripts using specific PASs.

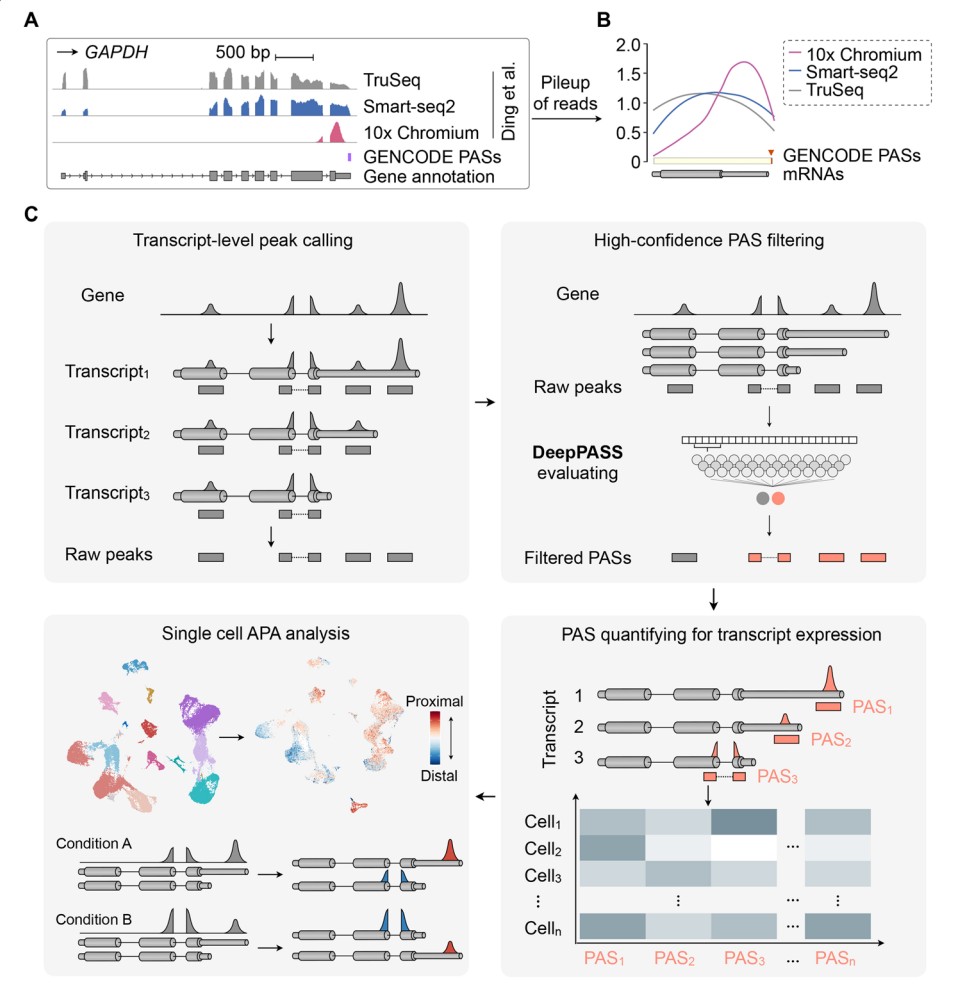

**Fig. 1** Developing SCAPTURE to identify cleavage and polyadenylation sites (PASs) from 3′ tag-based scRNA-seq. **a** Comparison of human PBMC transcriptome profiling with different deep sequencing datasets. Wiggle tracks show an enrichment of 10x Chromium reads (rose) at the 3′ end of the *GAPDH* gene locus, close to its known PAS (GENCODE), while reads of TruSeq RNA-seq (gray) and Smart-seq2 (dark blue) cover the whole gene body. Data were retrieved from published PBMC TruSeq RNA-seq, Smart-seq2, and 10x Chromium [17]. **b** Distribution of deep sequencing reads on mRNA genes. Pileup of deep sequencing reads from the same published datasets (**a**) indicates enrichment of 10x Chromium reads (rose) at 3′ ends of genes, compared to coverage of gene bodies by TruSeq RNA-seq (gray) and Smart-seq2 (dark blue) data. The distribution of PASs on mRNA genes were annotated in GENCODE. **c** Schematic of a stepwise SCAPTURE pipeline for single-cell PAS calling, filtering, transcript calculating, and APA analyzing. Top left, calling peaks from 3′ tag-based scRNA-seq (the first step). Top right, identifying high-confidence PASs with an embedded deep learning neural network DeepPASS (the second step). Bottom right, quantifying PASs to represent transcript expression at a single-cell resolution (the third step). Bottom left, applying SCAPTURE to APA analysis at single-cell level (the fourth step). See "Methods" section for details

## Results

### Development of SCAPTURE for PAS analysis with 3′ tag-based scRNA-seq data

To utilize 3′ tag-based scRNA-seq data for PAS analysis, we developed a stepwise computational pipeline named *sc*RNA-seq *a*nalysis for *P*AS-based *t*ranscript expression *u*sed to *re*fine cell identities (SCAPTURE, Fig. 1c; "Methods" section). Briefly, SCAPTURE takes aligned bam files as input to call peaks that are close to PASs of genes (Fig. 1c, top left). These called peaks are then evaluated by an embedded deep learning method to select high-confidence PASs (Fig. 1c, top right). Next, selected PASs are quantified by UMI-tools [18] to indicate expression of transcripts according to their distinct PAS usage (Fig. 1c, bottom right). Finally, SCAPTURE identifies altered PAS usage and transcript expression among different cell types/conditions at the single-cell level (Fig. 1c, bottom left). Two features of SCAPTURE are of note. First, a deep learning neural network, named DeepPASS, is embedded in the SCAPTRUE pipeline. It is trained by selecting sequences through shifting around known PASs with stringent filtering (called stringent PASs for simplicity, "Methods" section), resulting in identification of high-confidence PASs (Fig. 2a; Additional file 1: Fig. S2A; Additional file 2: Table S1; "Methods" section). Briefly, a predicted probability ranging between 0 and 1 is obtained by the DeepPASS model for any given PAS candidate, followed by classification into a positive site group (predicted probability > 0.5) or a negative site group (predicted probability ≤ 0.5) (Fig. 2b, "Methods" section). Compared with commonly used methods that are based on fixed sequences for feature extraction (termed DeepPASS-fixed here), DeepPASS achieved a higher area under curve (AUC) value with the training set (randomly selected from 90% of stringent PASs, "Methods" section) (Fig. 2c). As a result, by combining a convolutional neural network (CNN) and a recurrent neural network (RNN) for data training, DeepPASS achieved an AUC over 0.99 when using the validation set (the remaining, independent 10% of stringent PASs, "Methods" section), substantially higher than previously reported methods, such as DeepPASTA [19] and APARENT [20] (Fig. 2d; "Methods" section). Notably, this sequence selection through shifting strategy in DeepPASS can not only generate a larger training set than using fixed sequences, boosting its accuracy in PAS, but also makes predictions less sensitive to positions, increasing its sensitivity (Fig. 2e). Profiling active motifs of 128 kernels from first convolutional layer of DeepPASS showed that it was able to capture key motifs of PASs. Top captured motifs included canonical poly(A) signal AAUAAA (Entropy: 4.76) and its variants located 25 bp upstream of the cleavage site, as well as downstream U-rich motifs and the typical CA nucleotides located at the cleavage site (Additional file 1: Fig. S2B), further suggesting that DeepPASS is able to identify high-confidence PASs.

Another key feature of SCAPTURE is that the quantitative information of identified PASs is used to represent expression of transcript isoforms with distinct PASs (Fig. 1c, bottom right). It is well known that, through the alternative polyadenylation (APA) mechanism, multiple transcripts with distinct PASs (called APA transcripts) can be produced from a single-gene locus [21–23], increasing the transcriptomic complexity of the genome. Differential expression of APA transcripts has been widely examined across different cell types, but rarely at the single-cell resolution. Since different APA transcripts harbor distinct PASs, quantified PAS values can be used to represent differential transcript expression (DTE) at given gene loci. Hence, the DTE information generated by SCAPTURE can help refine cell identity analysis beyond gene expression (Fig.

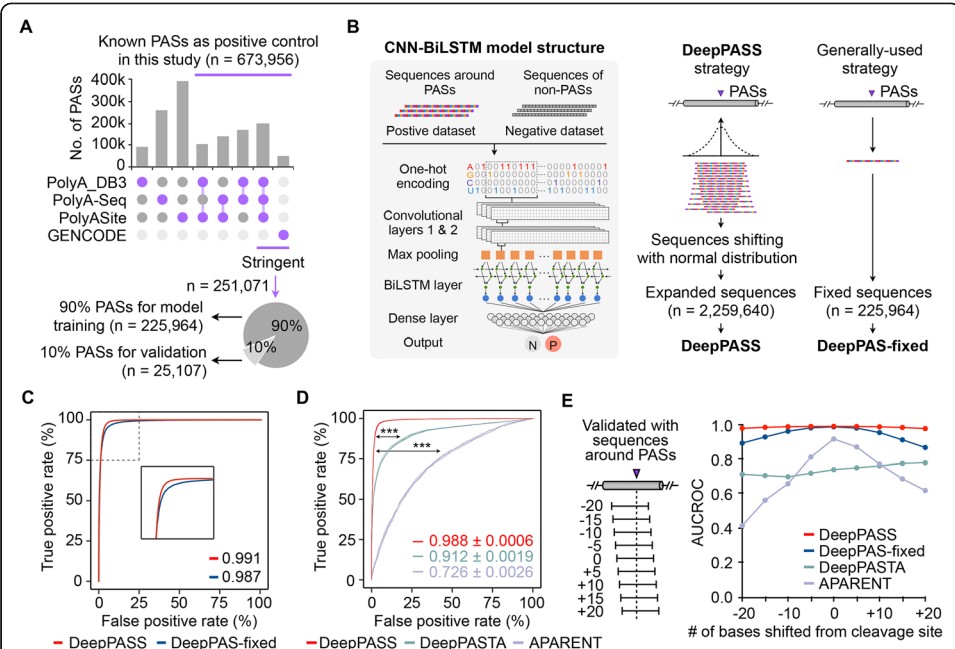

**Fig. 2** Constructing the embedded DeepPASS model for position-insensitive prediction of PASs. **a** Construction of known PAS set and stringent PAS set. Previously reported PASs in at least two databases of PolyA_DB3, PolyA-seq, and PolyASite (v2.0) or in the manually examined GENCODE annotation were combined to achieve known PAS set. Stringent PAS set was further constructed from known PASs that are annotated in all three databases or in the GENCODE annotation and was split with a 9:1 ratio between a model training set and an independent validation set for DeepPASS and DeepPAS-fixed models. **b** Schematic of DeepPASS construction and evaluation. Left, data processing strategy and model architecture. Middle, a sequence shifting strategy around stringent PASs was applied to construct positive training set for establishing DeepPASS model. Right, the generally used strategy with fixed sequences around stringent PASs for DeepPAS-fixed model. See "Methods" section for details. **c** The ROC curves of DeepPASS and DeepPAS-fixed to indicate their training performance. AUC values of DeepPASS (red) and DeepPAS-fixed (blue) were shown in plot. **d** The ROC curves of DeepPASS, DeepPASTA, and APARENT on the validation set. AUC values of DeepPASS (red), DeepPASTA (green), and APARENT (purple) were shown in plot from five independent validation repeats with very low standard errors. ***$p < 0.001$, statistical significance was assessed by Student's *t* test. See "Methods" section for details. **e** Position-insensitive prediction of PASs with DeepPASS model. To assess positional tolerance of different models, 200-bp sequences shifting around the PASs in validation set with 5 bp stride were used to test accuracy of each model. The percentage represents true positive rate of each condition. DeepPASS (red) is more tolerant than DeepPAS-fixed (blue) and previously reported DeepPASTA (green) and APARENT (purple) models

1c, bottom right; "Methods" section). By contrast, typical scRNA-seq tools perform cell clustering by using the DGE information only [3].

## SCAPTURE for exonic PAS analysis with human PBMC scRNA-seq data

We applied SCAPTURE to profile PASs from publicly available scRNA-seq datasets of human PBMCs based on the 10x Genomics platform (https://support.10xgenomics. com/single-cell-gene-expression/datasets) (six datasets in total; Additional file 1: Fig. S3A, B). Of the 83,390 raw peaks called in exons, 35,378 high-confidence PASs (selected by DeepPASS with predicted probability > 0.5) were identified by SCAPTURE (Fig. 3a, left; Additional file 1: Fig. S4A; Additional file 3: Table S2). Among them, 29,664 (83.8%) peaks overlapped with known PASs, which were named overlapped exonic PASs. The rest were named non-overlapped exonic PASs (Fig. 3a, right; Additional

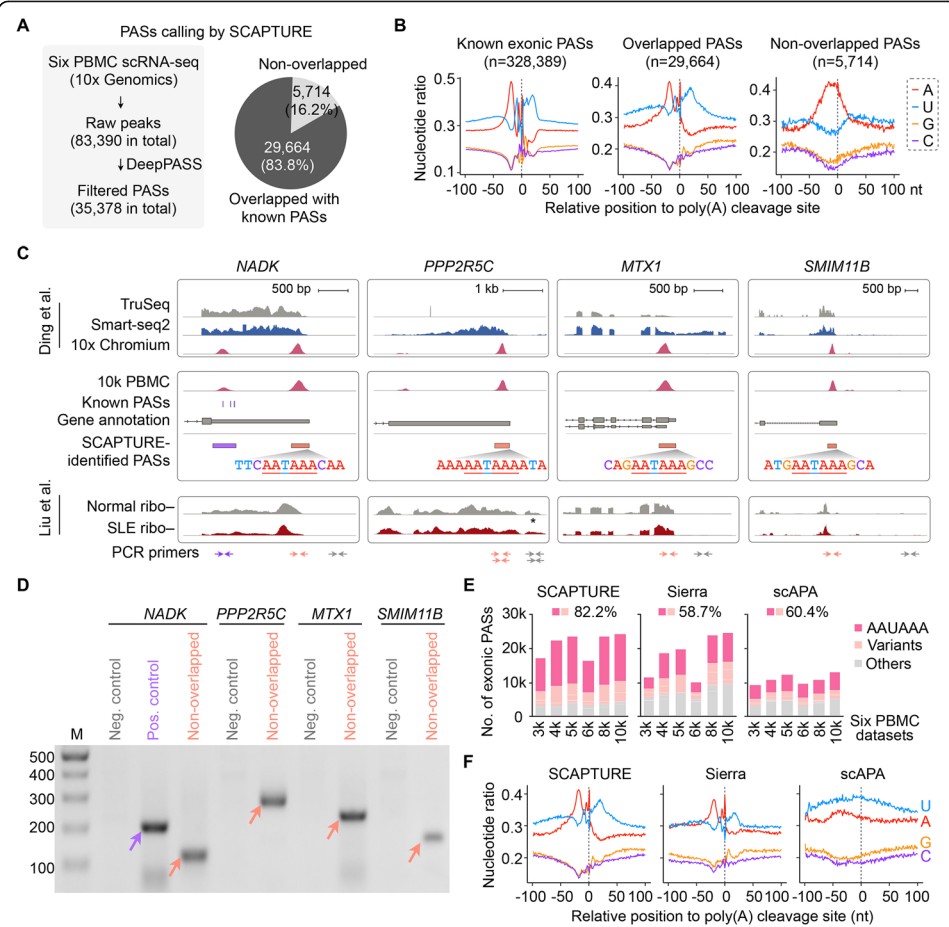

**Fig. 3** Applying SCAPTURE pipeline to study PASs in human PBMC datasets. **a** Application of SCAPTURE to identify exonic PASs from 3′ tag-based scRNA-seq datasets. Left, statistics of called peaks and identified PASs in six PBMC scRNA-seq datasets from 10x Genomics. Right, comparison of SCAPTURE-identified exonic PASs with known exonic PASs. Overlapped or non-overlapped exonic PASs identified by SCAPTURE were analyzed with known PAS annotation shown in Fig. 2a. **b** Nucleotide distribution of sequences at known (left), overlapped (middle), and non-overlapped (right) exonic PASs. Upstream (–) 100 bp to downstream (+) 100 bp sequences of PASs were analyzed. **c** Validation of SCAPTURE-identified non-overlapped PASs. Four such cases (salmon rectangle) in the *NADK*, *PPP2R5C*, *MTX1*, and *SMIM11B* gene loci highlighted with their polyadenylation signals (AAUAAA motif) and further designed for validation by PCR as previously reported [26]. A SCAPTURE-identified PAS that was previously annotated in the *NADK* gene locus (purple rectangle) was shown as a positive control. Top panel, TruSeq RNA-seq (gray), Smart-seq2 (dark blue), and 10x Chromium (rose) scRNA-seq from published datasets [17]. Middle panel, wiggle tracks from one of 10x Genomics PBMC scRNA-seq datasets (the 10 k PBMC dataset) were indicated. Bottom panel, Ribo–RNA-seq datasets of bulk cells from normal (gray) and SLE patient (dark red) PBMC samples [25]. PCR primers were shown in the bottom for experimental validation. See "Methods" section for details. Asterisk, non-specific RNA-seq signals due to multiple mapping. **d** Validation of SCAPTURE-identified non-overlapped exonic PASs. Corresponding PCR products of these non-overlapped exonic PASs (salmon arrow) were shown with correct sizes. One overlapped exonic PAS (purple arrow) that was also identified by SCAPTURE in the *NADK* gene locus was shown as the positive control. As negative controls, no PCR products from downstream regions of these SCAPTURE-identified non-overlapped PASs were detected. Mixed PBMC RNA samples from SLE patients previously examined by Liu et al. [25] were used for this validation. Of note, additional nested primers were designed to validate the SCAPTURE-identified PAS in the *PPP2R5C* locus. **e** Comparison of signature poly(A) signal motifs from exonic PASs identified by SCAPTURE or by two other recently reported pipelines, Sierra [28] and scAPA [29]. **f** Nucleotide distribution of sequences around exonic PASs identified by SCAPTURE, Sierra, or scAPA in six PBMC datasets from 10x Genomics. Upstream (–) 100 bp to downstream (+) 100 bp sequences of PASs were analyzed

file 1: Fig. S4B). To examine the accuracy of predicted PASs, we profiled their surrounding nucleotides [24]. As shown in Fig. 3b, a typical nucleotide distribution profile was observed in the overlapped exonic PASs, similar to that of known exonic PASs. For non-overlapped exonic PASs, although they had a similar nucleotide distribution profile, downstream U-rich sequences were less enriched (Fig. 3b). Nevertheless, while slightly lower than that of overlapped ones (81.4%), 68.4% of non-overlapped exonic PASs harbored canonical AAUAAA or its variants, which was similar to that of known PASs (67.2%) but much higher than that of SCAPTURE-rejected ones (36.7%; Additional file 1: Fig. S4C).

Of note, most of SCAPTURE-identified non-overlapped exonic PASs located near the 3′ ends of transcripts, similar to those of known and overlapped PASs (Additional file 1: Fig. S4D). Four of SCAPTURE-identified non-overlapped exonic PASs located at *NADK*, *PPP2R5C*, *MTX1*, and *SMIM11B* gene loci (Fig. 3c), with DeepPASS predicted probabilities of 0.85~0.99 and exhibiting relatively high scRNA-seq peaks, were chosen for further validation. Their authenticity was supported by TruSeq and/or Smart-seq2 datasets based on different PBMC samples [17], including those from autoimmune disease systemic lupus erythematosus (SLE) patients [25] (Fig. 3c). In addition, we carried out RT-PCR validation of these non-overlapped exonic PASs with mixed PBMC RNA samples from human SLE patients [25], using a similar strategy for PAS validation to that reported previously [26]. With designed primer sets (Additional file 4: Table S3) located within scRNA-seq peaks, all four of these SCAPTURE-identified non-overlapped (distal) exonic PASs were confirmed by PCR products with correct sizes (Fig. 3d). As a positive control, a reported (proximal) PAS in the last exon of *NADK* gene was also confirmed (Fig. 3d). By contrast, PCR signals from downstream regions of these SCAPTURE-identified non-overlapped exonic PASs were barely detectable (Fig. 3d). Together, these results suggest that many SCAPTURE-identified non-overlapped PASs might be bona fide PASs (Fig. 3c).

### SCAPTURE for intronic PAS analysis

PAS profiling often suffers from false positives that are generated from internal priming of oligo(dT) at A-rich sequences, especially in intronic regions [27, 28]. To evaluate whether our DeepPASS model could deal with the internal priming problem, we used intergenic sequences with consecutive adenines that were far away from annotated PASs as pseudo internal priming sites for examination. Strikingly, DeepPASS could efficiently identify these sites as true negative sites (> 94.9%; Additional file 1: Fig. S5A). Additionally, DeepPASS successfully identified internal priming sites with different lengths of consecutive adenines (Additional file 1: Fig. S5A), supporting its robustness in addressing the internal priming issue. With the stringent calling by SCAPTURE, we detected 16,082 intronic PASs from the 10x Genomics PBMC samples (Additional file 1: Fig. S5B; Additional file 3: Table S2). Although ~ 18.6% of SCAPTURE-identified intronic PASs overlapped with known intronic sites, this number is still much higher than those (~ 2.1% and 2.0%, respectively) of Sierra- and scAPA-identified intronic PASs (Additional file 1: Fig. S5C). Peak quantification indicated that intronic PAS transcripts were expressed at much lower levels compared to those of exonic PAS ones (Additional file 1: Fig. S5D), which may explain why these sites were not annotated in

public databases. Nevertheless, both overlapped and non-overlapped intronic PASs showed comparable occurrences of AAUAAA and its variants to that of known intronic PASs (78.5% and 82.7% *vs* 78.0%, respectively; Additional file 1: Fig. S5E), which were much higher than that of SCAPTURE-rejected ones (51.8%; Additional file 1: Fig. S5E).

Using RT-PCR [26], we further confirmed four SCAPTURE-identified, non-overlapped intronic PASs (Additional file 1: Fig. S5F), all of which passed the evaluation by the DeepPASS model with predicted probabilities ranging from 0.58 to 0.92 and had no more than six consecutive adenines in their surrounding regions. With nested PCR primers (Additional file 1: Fig. S5G), all these SCAPTURE-identified non-overlapped intronic PASs could be successfully confirmed by corresponding PCR products with correct sizes (Additional file 1: Fig. S5H). In contrast, no PCR signals could be detected from their downstream regions. This observation thus indicates that some of SCAPTURE-identified non-overlapped intronic PASs are true positives. In spite of this, due to the challenge in calling intronic PAS, we advocate that additional attention should be paid in the study of intronic PASs, perhaps with more specifically developed methodologies.

### Comparison of SCAPTURE with other methods

Two methods, Sierra [28] and scAPA [29], were recently reported to analyze PASs from scRNA-seq data (Additional file 1: Fig. S6A). For the same human PBMC scRNA-seq datasets based on the 10x Genomics platform, SCAPTURE identified more exonic PASs (mean = 21,274) than did Sierra and scAPA (mean = 18,166 or 11,046, respectively; Fig. 3e; Additional files 5, 6, 7 and 8: Table S4-7). To further confirm the reliability of SCAPTURE, we selected for validation five additional non-overlapped exonic PASs that were identified by SCAPTURE (with a predicted probability of 0.58~0.88 and relatively high peaks, Additional file 1: Fig. S6B) but not by Sierra or scAPA (Additional file 4: Table S3). By using a similar PCR strategy for PAS validation (Fig. 3d; Additional file 1: Fig. S5H) with nested PCR primers (Additional file 1: Fig. S6C), all these non-overlapped exonic PASs were confirmed (Additional file 1: Fig. S6D). In addition, the SCAPTURE-identified exonic PASs displayed higher average frequencies of AAUAAA and its variant than those by Sierra and scAPA (82.2% by SCAPTURE *vs* 58.7% by Sierra or 60.4% by scAPA; Fig. 3e). Moreover, SCAPTURE and Sierra performed better on identified exonic PASs than did scAPA, as evidenced by nucleotide distributions (comparing Fig. 3f with Fig. 3b). The motif preference (Fig. 3e) and nucleotide distribution (Fig. 3f) analyses both supported higher accuracy of SCAPTURE than the other two pipelines. Finally, although SCAPTURE identified fewer intronic PAS candidates than Sierra or scAPA (Additional files 5, 6, 7 and 8: Table S4-7), those SCAPTURE-identified intronic PASs exhibited higher average frequencies of AAUAAA and its variant than those by Sierra and scAPA (83.2% by SCAPTURE *vs* 50.8% by Sierra or 28.3% by scAPA; Additional file 1: Fig. S6E). Nucleotide distribution analysis also supported SCAPTURE's better performance in detecting intronic PASs than Sierra and scAPA, because the latter two both identified PASs with downstream A-rich sequences, a hallmark of the internal priming issue (Additional file 1: Fig. S6F). Taken together, our data indicate that SCAPTURE captures PAS information from 3′ tag-based scRNA-seq data with high sensitivity and accuracy.

## SCAPTURE has high sensitivity and specificity

To calculate the sensitivity and specificity of SCAPTURE, 34,108 PAS peaks called by the first step of SCAPTURE with expression of RPM ≥ 1 (reads calculated from Cell Ranger mapped BAM files within identified peak regions of PASs, "Methods" section) from examined PBMC samples were used for two distinct classifications (Additional file 1: Fig. S7A). On the one hand, 34,108 expressed peaks were filtered through the Deep-PASS model to obtain "predicted" positive PASs (with predicted probability score > 0.5) or "predicted" negative PASs (with predicted probability score ≤ 0.5). On the other hand, with a previously published method [30] to quantify and evaluate PASs by enrichment scores (Additional file 1: Fig. S7B), 34,108 expressed peaks were also divided into two groups as "actual" positive PASs (enrichment score ≥ 2) or "actual" negative PASs (enrichment score < 2). With these predicted and actual sets, the sensitivity and specificity of SCAPTURE were calculated to be 0.69 and 0.66, respectively, with an $F_1$ score of 0.74 (Additional file 1: Fig. S7C). Similar sensitivity and specificity values of SCAPTURE were obtained when different enrichment scores (≥ 1.5, 1.75, 2, 2.25, or 2.5, respectively) as cutoffs were used to divide "actual" positive and negative PAS sites (Additional file 1: Fig. S7C).

To further compare SCAPTURE with Sierra and scAPA in recapturing stringent exonic PASs in databases. In both training and validation sets, ~ 71% and 70% of expressed stringent exonic PASs in examined PBMCs with RPM ≥ 1 (reads calculated from Cell Ranger mapped BAM files within upstream 400 bp of stringent PASs, "Methods" section) could be respectively identified by SCAPTURE, indicating that its sensitivity (Additional file 1: Fig. S7D and S7E) is similar to that calculated by the enrichment score (Additional file 1: Fig. S7C). By contrast, the sensitivity scores of Sierra and scAPA were about 60% and 47%, respectively (Additional file 1: Fig. S7D and S7E). These results together indicate that SCAPTURE has a higher sensitivity in PAS calling.

## SCAPTURE quantifies APA transcripts for refined cell identity analysis

We next reasoned that quantified PAS values from 3′ tag-based scRNA-seq could represent the expression of corresponding APA transcripts (Additional file 1: Fig. S8) and thus facilitate subsequent single-cell clustering. Different to the commonly used DGE-based methods, the DTE data from SCAPTURE embody multiple APA transcript expression information from single-gene loci, due to different PAS usages (Fig. 4a). Importantly, over half of the genes detected in human PBMC datasets from 10x Genomics expressed two or more APA transcripts (Fig. 4b, c) and, among the top 2000 high variable PAS-based features, ~ 60.0% of them expressed APA transcripts (Fig. 4d). Hence, DTE information provides an opportunity to further refine cell identities, beyond the informion from DGE data.

Parallel analyses demonstrated that SCAPTURE analysis using DTE had a similar power on sample integration, dimension reduction, and unsupervised clustering to the canonical Seurat protocol [31] using DGE (Additional file 1: Fig. S9A-C; Additional file 9: Table S8). However, while both approaches could accurately identify major cell types, the numbers of each cell type were different (Fig. 4e). More cells were unassigned by Seurat than by SCAPTURE (237 vs 172; Fig. 4e). In addition, unassigned cells in DTE analysis were clustered away from all other cell types (Fig. 4e, left), suggesting that they

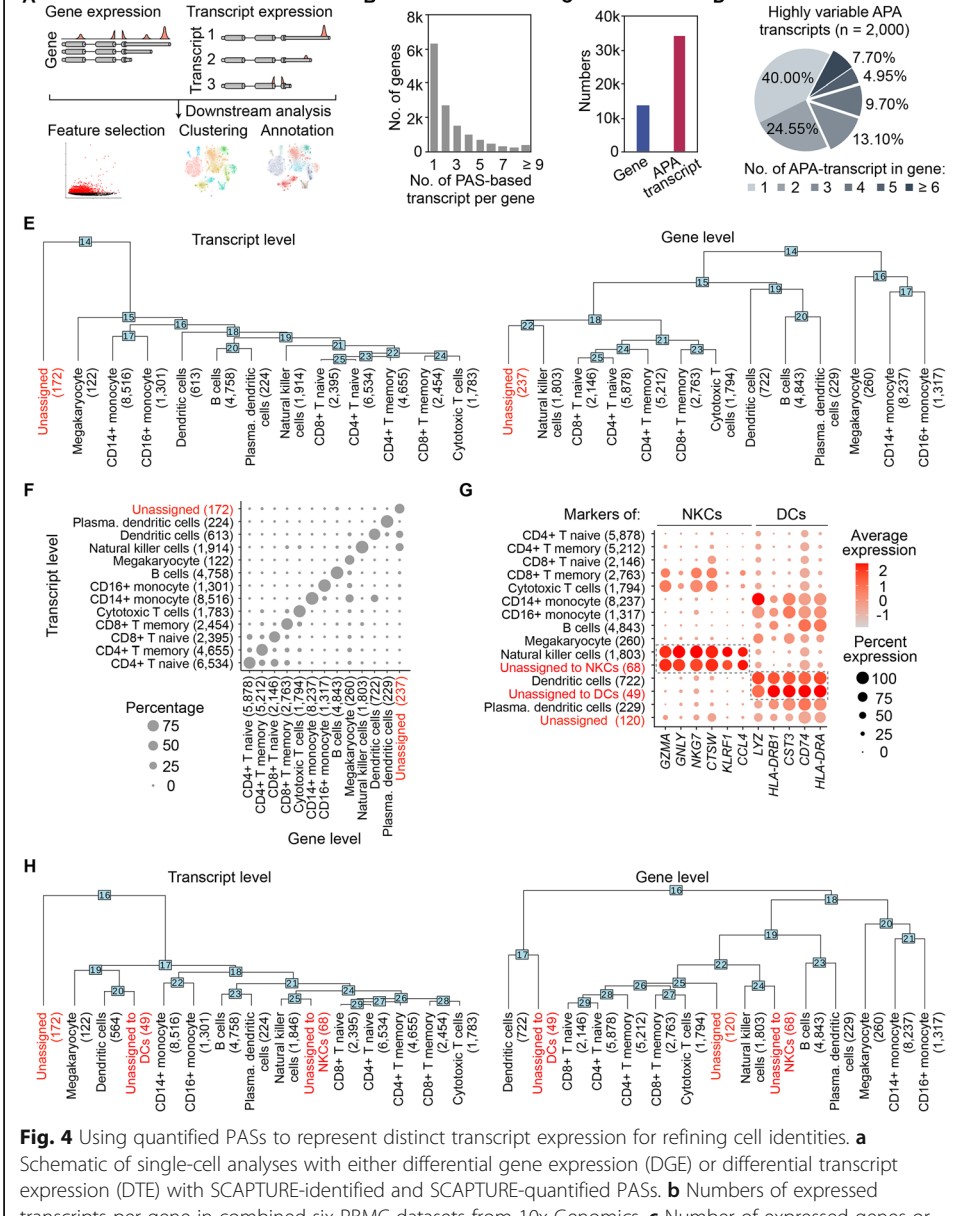

**Fig. 4** Using quantified PASs to represent distinct transcript expression for refining cell identities. **a** Schematic of single-cell analyses with either differential gene expression (DGE) or differential transcript expression (DTE) with SCAPTURE-identified and SCAPTURE-quantified PASs. **b** Numbers of expressed transcripts per gene in combined six PBMC datasets from 10x Genomics. **c** Number of expressed genes or transcripts in combined six PBMC datasets from 10x Genomics. **d** Gene distribution of top 2000 highly variable PAS-based transcript features. Of note, the majority of these top variable features are associated with genes expressing APA transcripts. **e** Phylogenetic analysis of cell types clustered by DTE with SCAPTURE (left) or by DGE with the conventional Seurat protocol [31] (right). **f** Cross comparison of cell types assigned by DTE or DGE. **g** Comparison of marker gene expression between DGE-assigned major cell types (black) and DGE-unassigned but DTE-assigned cells (red). Of note, among the DGE-unassigned but DTE-assigned cells, some showed similar marker gene expression patterns as natural killer cells (NKCs; 68 out of 237) or dendritic cells (DCs; 48 out of 237). **h** Phylogenetic re-analysis of cell types clustered by DTE (left) or by DGE (right). DGE-unassigned but DTE-assigned cells are highlighted in both clusters (red)

were truly distinct populations. By contrast, unassigned cells were clustered close to natural killer cells (NKCs) in DGE analysis (Fig. 4e, right), suggesting that cell clustering by DGE might not resolve cell identities as well. Direct comparisons showed that many of the unassigned cells in DGE analysis could be clustered by DTE to known cell types, such as dendritic cells (DCs) or NKCs (Fig. 4f). Moreover, the similarity of these

unassigned cells by DGE to their corresponding NKCs or DCs was supported by their comparable patterns of marker gene expression (Additional file 10: Table S9), which were then manually sub-grouped to "unassigned to NKCs" and "unassigned to DCs," respectively (Fig. 4 g; Additional file 1: Fig. S9D, "Methods" section). As expected, these unassigned cells could be re-clustered close to DCs or NKCs after manual subgrouping (Fig. 4h). Together, these results highlight the advantage of transcript-level analysis offered by SCAPTURE in single-cell studies.

### SCAPTURE identifies altered PAS usages in patients infected with SARS-CoV-2

By comparing proximal and distal PAS usages in the benchmark PBMC scRNA-seq data, we found less proximal PAS usage in monocytes compared to other major types of immune cells, including DCs, T cells, B cells, and NKCs (Fig. 5a). Next, we applied SCAPTURE to PBMC scRNA-seq data from healthy individuals or COVID-19 patients infected with SARS-CoV-2 [32] (Fig. 5b). In total, nine PBMC scRNA-seq samples were from four severe COVID-19 patients and five healthy individuals ("Methods" section) [32] (Fig. 5c). The performance of SCAPTURE, including batch effect removal, dimension reduction, and unsupervised clustering, was consistent with regular DGE-based analyses (Additional file 1: Fig. S10A; Additional file 11: Table S10). Similarly, less proximal PAS usage in monocytes of PBMCs was detected in both healthy individuals and severe COVID-19 patients (Fig. 5d, e). The differential PAS usages between monocytes and DCs at some gene loci are shown in Fig. 5f. These genes have known functions in activation of innate/adaptive immune response, antigen processing/presentation via MHC, or dendritic spine development, suggesting distinct APA functions in different types of immune cells.

Strikingly, compared to those in healthy individuals, a general preference of proximal PAS usage, indicating 3′UTR shortening, was detected in all major PBMC cells of severe COVID-19 patients (Fig. 5g). This result was in line with previous bulk cell data indicating preferential activation of proximal PASs in PBMCs upon infections [33, 34]. Therefore, 3′UTR shortening in PMBCs may be a common phenomenon after viral infection, including SARS-CoV-2. Moreover, GO analysis of altered proximal PAS usage between healthy and COVID-19 samples showed functional enrichment of genes involved in immune responses, Fc receptor signal pathway, B cell activation, etc. (Additional file 1: Fig. S10B; Additional file 12: Table S11). Additional profiling of immune response-related genes highlighted the preferential proximal PAS usage of immunoglobulin genes, such as *IGHM*, *IGHG1*, *IGHG3*, and *IGHA2*, especially in B cells and plasma cells of COVID-19 samples (Fig. 5h; Additional file 1: Fig. S10C). It has been previously reported that the usage of proximal PAS in immunoglobulin heavy chain genes, such as *IGHM* [35, 36], is important to their protein functions. For example, the *IGHM* isoform using the proximal PAS encodes a secreted form of IgM antibody, whereas the isoform using the distal PAS encodes a membrane-bound form of IgM antibody [37]. Thus, the preferential proximal PAS usage in immunoglobulin heavy chain genes in COVID-19 patients as revealed by SCAPTURE is in good agreement with the general APA site switch after viral infection. Of note, while DGE analysis also indicated expression changes of these immunoglobulin heavy chain genes in COVID-19 patients (mostly enhanced, Additional file 1: Fig. S10D), DTE analysis by

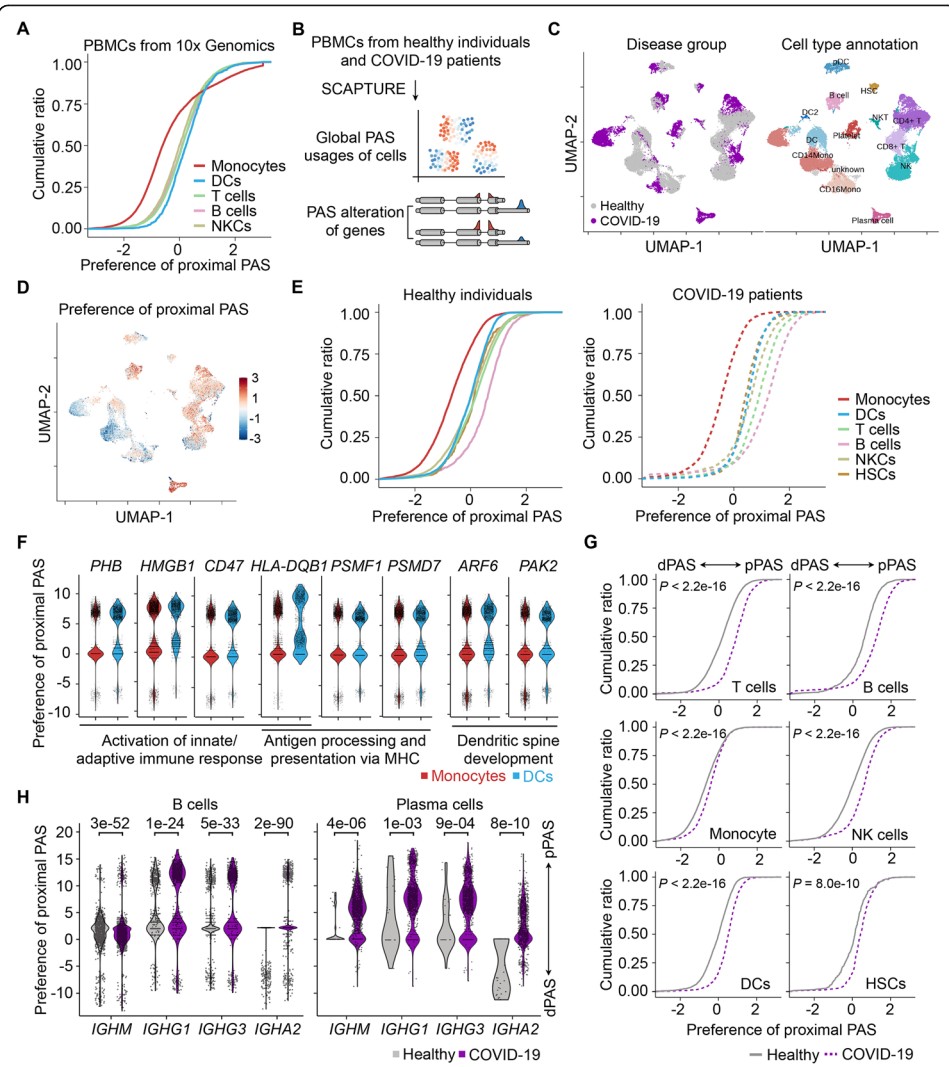

**Fig. 5** Identifying altered PAS usages upon SARS-CoV-2 infection. **a** Less proximal PAS usage in monocytes than other immune cell types, such as DCs, in six PBMC scRNA-seq datasets from 10x Genomics. **b** Schematic of APA analysis by SCAPTURE to compare PAS usage changes between healthy individuals and severe COVID-19 patients with SARS-CoV-2 infection at single-cell level. See "Methods" section for details. **c** UMAP plots showing DTE-based single-cell clustering of PBMC scRNA-seq datasets from healthy individuals and COVID-19 patients by SCAPTURE. Left, single cells labeled with disease group. Right, single cells labeled with annotated cell type. **d** UMAP plots showing global preference of proximal PAS usage in healthy individuals and COVID-19 patients at single-cell level. "Methods" section for details. **e** Proximal PAS usage in immune cell types from PBMC scRNA-seq datasets of healthy individuals (left) and COVID-19 (right) patients. **f** Genes with altered PAS usage between monocytes and DCs. Examples of known genes with reported functions, such as activation of innate/adaptive immune response, antigen processing/presentation via MHC, or dendritic spine development, were shown. **g** Global preference of proximal PAS usage in different immune cell types from COVID-19 samples. Statistical significance was assessed by Kolmogorov-Smirnov test. **h** Preferential proximal PAS usage of some immunoglobulin genes in B cells (left) and plasma cells (right) of COVID-19 samples. Statistical significance was assessed with Seurat ("Methods" section)

SCAPTURE provided a more refined picture of expression changes of these genes at the transcript isoform level (Fig. 5h).

## Discussion and conclusion

In this study, we report a stepwise computational pipeline SCAPTURE for de novo identifying PASs from 3′ tag-based scRNA-seq datasets (Fig. 1). With an embedded deep learning neural network DeepPASS, which extracts features from sequences shifting around known PASs (Fig. 2), SCAPTURE efficiently identifies PASs, as shown in using a benchmark 3′ tag-based scRNA-seq dataset of normal PBMCs from 10x Genomics, including thousands of novel PAS sites (Fig. 3). Notably, these novel PASs harbor a similar nucleotide distribution profile to known PASs (Fig. 3e, f). With a previously developed PCR strategy for PAS validation [26], we corroborated a few of these novel exonic and intronic PASs (Fig. 3d; Additional file 1: Fig. S5H and S6D). Compared to other models with fixed sequences for training or other reported Deep-PASTA [19] and APARENT [20] methods, the DeepPASS method used in SCAPTURE showed higher AUCs for position-insensitive prediction of PASs (Fig. 2c–e). SCAPTURE also showed higher sensitivity and accuracy than Sierra and scAPA pipelines (Additional file 1: Fig. S6 and S7). On the other hand, while more reliable than Sierra and scAPA (Additional file 1: Fig. S5), SCAPTURE could suffer from false positives in intronic PAS identification due to their low expression [27, 28]. Thus, cautions are needed when calling intronic PASs from scRNA-seq data.

Another feature of SCAPTURE is using quantified PAS values from scRNA-seq data to represent the expression of APA transcripts, providing additional molecular signatures of gene expression for refined cell identity analysis (Fig. 4). Differential PAS usage in different cell types was observed in immune cells of PBMCs from either normal/healthy samples or severe COVID-19 patients (Fig. 5a, e). In addition, compared to those in healthy individuals, differential PAS usage, especially a general preference toward proximal PAS usage leading to shortened 3′UTRs, was detected in all major PBMC cell types of COVID-19 patients (Fig. 5g). Consistently, the preferential proximal PAS usage of some immunoglobulin heavy chain genes in B cells and plasma cells from COVID-19 patients (Fig. 5h; Additional file 1: Fig. S10C) is in good agreement with activated immune response upon SARS-CoV-2 infection [37]. These results further support the notion that PAS usage is substantially regulated upon infection [33, 34], leading to differential expression of APA transcripts. This finding is consistent with recent studies indicating the roles of secreted antibodies from immunoglobulin genes in neutralizing SARS-CoV-2 virus [38–40].

In summary, we present the SCAPTURE pipeline to identify PASs in single cells. SCAPTURE detects PASs precisely with the incorporation of a deep learning model, DeepPASS. In addition, we show SCAPTURE quantifies PASs from scRNA-seq datasets, which aids in refinement of single-cell clustering by using APA transcript expression. Finally, we profile different PAS usages between healthy individuals and COVID-19 patients and identify the tendency of proximal PAS usage in many immune response-related genes upon SARS-CoV-2 infection. We envisage that, with fast-growing scRNA-seq datasets, SCAPTURE can produce more precise PAS atlas at the single-cell levels, further refining single-cell clustering and shedding light on APA functions under diverse physiological and pathological conditions.

## Methods

### Collection of reported PASs from different annotations to obtain known PASs

Four publicly available databases of PASs were used in this study, including manually annotated poly(A) sites in GENCODE (v35) ($n$ = 49,942) and other three databases from published literatures: PolyA_DB3 [41] ($n$ = 290,051), PolyA-seq [42] ($n$ = 514,262) and PolyASite [43] (v2.0, $n$ = 569,005). To obtain annotated PASs as positive controls, previously reported PASs in at least two databases of PolyA_DB3, PolyA-seq, and Poly-ASite (v2.0) or in the GENCODE annotation were combined to get 673,956 known PASs (Additional file 1: Fig. S2A, left), which were used to compare with pipeline-called PASs from scRNA-seq datasets in this study. If any of these 673,956 known PASs were found to be located in the regions between upstream 50 bp to downstream 25 bp of 3′ ends of called PAS peaks, such called PAS peaks were treated as overlapped sites with known PASs. Of note, among the 673,956 known PASs, 251,071 of them were identified in all three databases, including PolyA_DB3 [41], PolyA-seq [42] and PolyA-Site [43], or GENCODE annotation, and were further classified as stringent PASs for the model training and validation (Additional file 1: Fig. S2A, right) (detailed below).

### Collection of published scRNA-seq and bulk cell RNA-seq data for analysis

Multiple types of RNA-seq datasets from independent resources were used in this study. A collection of bulk cell RNA-seq (library prepared with Illumina TruSeq), full-length scRNA-seq (Smart-seq2, specifically) and 3′ tag-based scRNA-seq (10x Chromium, specifically) datasets of human PBMCs, was obtained from a published literature by Ding et al. [17], individually called as TruSeq, Smart-seq2, or 10x Chromium in this study. Another collection of six 10x Chromium datasets from human PBMCs, was downloaded from 10x Genomics company website (https://support.10xgenomics.com/single-cell-gene-expression/datasets). According to different single-cell numbers, they were individually named as 3 k, 4 k, 5 k, 6 k, 8 k or 10 k PBMC sample. Wiggle tracks of 10 k PBMC sample from 10x Genomics were used as examples to show the results analyzed by SCAPTURE pipeline (Fig. 3c; Additional file 1: Fig. S3B-C, S5F, S6B and S8). The third collection of bulk cell RNA-seq datasets were generated with ribosomal RNA-depleted RNAs of monocytes, B cells or T cells from PBMCs from SLE patients or normal donors [25]. In the current study, RNA-seq reads of monocytes, B cells and T cells from SLE patients or normal donors were pooled together for visualization. The last collection related to COVID-19 study including PBMC scRNA-seq datasets from five healthy individuals and four severe COVID-19 patients [32].

### Construction of SCAPTURE method

A stepwise computational method, SCAPTURE, was developed to capture PAS information from scRNA-seq data, and quantify alternatively polyadenylated transcripts for refining cell identities and studying APA regulation. SCAPTURE consists of three major steps to call peaks, to evaluate them for high-confidence PAS candidates and to finally quantify PASs. Since different transcripts harbor distinct PASs, quantified PAS values can be applied to represent differential transcript expression (DTE) at given gene loci to refine single-cell identities and perform APA analysis.

### Step 1: peak calling at the transcript level

Aligned BAM file, usually generated from Cell Ranger [13] (v4.0), was used as input to call peaks for subsequent PAS evaluation. Giving the fact of highly variable gene expression in scRNA-seq datasets, the relatively low expressed genes would be overwhelmed due to insufficient saturation or putatively rare cell types. To achieve high sensitivity, transcript-level peak calling was applied to detect peak signal at transcript level for a given gene locus using HOMER *findPeaks* (parameter: -size 400 -minDist 10 -strand separate -F 0 -L 0 -C 0 -gsize -tagThreshold). It has been reported that the typical read coverage of single PAS was a normal distribution-like curve and read coverage of adjacent PASs would overlap together [29]. As such, normality test and additional multimodality test were performed to assess peak signal of single PAS or adjacent PASs, respectively. After that, regions of called peaks with ≥ 50% overlapping from the same gene loci were aggregated and only the maximumly expressed peak was remained for subsequent analyses. After aggregation, read counts of peaks were quantified and low coverage peaks (read count percentage ≤ 1% among all peaks in the same gene loci) were removed. Additionally, for intronic PASs, we removed PASs with consecutive adenines ≥ 8 to avoid intronic internal priming sites.

### Step 2: PAS evaluation with an embedded deep learning neural network

Raw peaks called from scRNA-seq read signals contain PASs, but also suffer from false positive sites that are likely generated by aberrant read augment or internal priming of oligo(dT) at A-rich sequences during library construction. To efficiently identify high-confidence PASs from these called peaks, a deep learning neural network that was trained by sequences shifting (thus referred to as DeepPASS) around randomly selected stringent PASs was developed and embedded to evaluate called peaks from 3′ tag-based scRNA-seq. Peaks with a positive prediction (predicted probability > 0.5 by Deep-PASS) were considered as high-confidence PASs in further analysis. See below for detailed DeepPASS development (detailed below).

### Step 3: quantifying PASs to represent expression of alternatively polyadenylated transcripts

SCAPTURE could assign PASs of a gene to multiple transcripts that are produced at the same gene locus through alternative polyadenylation. Additionally, the width of peaks from 10x Chromium scRNA-seq is about 400 bp, longer than the average length of human exons (262 bp, GENCODE v34 hg38) and mouse exons (287 bp, GENCODE vM25 mm10), suggesting the called PAS peaks can span multiple exons. As such, SCAPTURE could partially decode the splicing information to assign peaks for corresponding transcripts in given gene loci. After transcript assignment, SCAPTURE-identified PASs were used to build a PAS-based GTF format annotation. To quantify PAS-based transcripts at a single-cell level, UMI-tools (v1.0.1) protocol was utilized to generated barcode count matrices. Briefly, reads from input aligned BAM file were re-assigned to PAS-based GTF annotation using featureCounts (parameter: -a GTF.file -t exon -g gene_id -M -O --largestOverlap -s 1 -R BAM --Rpath). The re-assigned BAM was used to calculate UMI counts at single-cell resolution for PAS-based transcript using UMI-tools *counts* (parameter: --extract-umi-method = tag --umi-tag = UB --cell-tag = CB --per-gene --gene-tag = XT --assigned-status-tag = XS --per-cell --wide-

format-cell-counts). The final count matrix of PAS-based transcript is in wide format and could be directly analyzed in downstream single-cell tools like Seurat.

## Comparison of PAS calling by SCAPTURE with other pipelines

Two previously reported pipelines, Sierra [28] and scAPA [29], were also applied for PAS calling from the same scRNA-seq human PBMC datasets from 10x Genomics with default parameters. The same gene annotation of GENCODE (v34 hg38) was also used to call PASs for Sierra. Since scAPA was developed with its own built-in reference of GENCODE (v33 hg19), genomic intervals of scAPA-called peaks were transferred to hg38 annotation using UCSC LiftOver for comparison.

## Development of DeepPASS embedded in SCAPTURE to evaluate PASs

A deep learning neural network, DeepPASS, was developed and embedded in SCAPTURE to evaluate called PAS peaks from 3′ tag-based scRNA-seq datasets for high-confidence PASs. A predicted probability ranging between 0 and 1 of a given candidate PAS is obtained by the DeepPASS model, resulting in its classification as a positive site (probability $> 0.5$) or a negative site (probability $\leq 0.5$). DeepPASS consists of four main steps to achieve a position-insensitive judgement on PASs.

### Step 1: training set construction

Stringent PASs were split with a 9:1 ratio between a model training set and an independent validation set (225,964 and 25,107, respectively) for the development of DeepPASS and DeepPAS-fixed. Specifically, 225,964 stringent PASs in the training set were used as input for DeepPASS and DeepPAS-fixed, while 25,107 stringent PASs in the validation set were used for comparison across different neural network models. In total, these 225,964 stringent PASs in the training set were used as positive sites to train DeepPASS and DeepPAS-fixed models in this study. For DeepPASS model, 200 bp sequences shifting around these PASs with a normal distribution ($\mu = 0$, $\sigma = 10$) were first extracted, followed by removal of redundant sequences to construct a positive training set ($n = 2{,}259{,}640$, $\sim 10$ sequences per stringent PAS, Fig. 2b). For DeepPAS-fixed model, 200-bp sequences with fixed positions around these PASs ($-100 \sim +100$ bp) were extracted, followed by removal of redundant sequences to construct its positive training set ($n = 225{,}964$). The same amounts of negative sequences, 2,259,640 for DeepPASS or 225,964 for DeepPAS-fixed, were constructed with 200 bp sequences randomly extracted in intergenic regions without overlapping with any of the annotated PASs in three databases or GENCODE. Of note, AUC values barely changed when different amounts of CDS/UTR sequences (up to 25%) were used as negative training sequences (data not shown).

### Step 2: model architecture

Both DeepPASS and DeepPAS-fixed are hybrid models of a convolutional neural network (CNN) and a recurrent neural network (RNN), which are constructed with TensorFlow v2.0.0 backend (http://tensorflow.org/) in Python v3.7.8. Both models can be summarized as:

$$\overline{O}_i = f^{Softmax} f^{Dense\_ReLU} f^{BiLSTM} f^{MaxPooling} f^{Conv\_ReLU} f^{Conv\_ReLU} \left( \overline{X}_i \right)$$

Each 200 bp sequence is transformed to One-Hot-Coded Matrix ($\overline{X}_i$), including binary vectors representing the presence or absence of 4 nucleotides: A (1, 0, 0, 0), T (0, 1, 0, 0), G (0, 0, 1, 0), C (0, 0, 0, 1), and labeled with 1 as being extracted around positive PASs or 0 as being extracted from negative non-poly(A) sites.

Given the characteristic *cis*-element of PASs, the CNN module is used to capture the core sequence around PASs with two convolutional layers utilizing Rectified Linear Unit (ReLU) activation function and subsequent max pooling layer. The first convolutional layer has 128 filters with 12-mer width and 4 channels. The second convolutional layer has 64 filters covering 128 output channels derived from the first layer, where the filters are 6-mer wide. The max pooling layer is used to subsample the signal from the previous layer by a stride of 4.

Considering the potential correlations of different *cis*-elements around PASs, a bi-directional long short-term memory (BiLSTM) layer is applied in a modified architecture of RNN. BiLSTM layer follows the CNN module to maintain the information in the context of DNA sequence appropriately. The BiLSTM layer has 128 units to process the output from the previous CNN module sequentially with two opposite directions and then passes the signal to a dense layer with 1024 hidden units and 0.3 dropout ratio. Then, dense layer connects with two output nodes ($\overline{O}_i$) using softmax activation function to predict the probabilities of two classes separately. Finally, DeepPASS predicts a certain class by choosing the maximal probability of each class from $\overline{O}_i$.

### Step 3: model training

DeepPASS and DeepPAS-fixed models were trained using corresponding training sequences (described in Step 1: training set construction) with a batch size of 5000 and 100 epochs. Early stopping was set with a patience of 10 rounds to terminate training process if no increased performance was observed. The model with best performance during training was kept at last.

### Step 4: investigating model learning features

To extract the motifs obtained by DeepPASS, the matrix filters in the first convolutional layer scanned through input DNA sequences and reported subsequences with maximal filter activation as reported strategy [44]. Then position probability matrix (PPM) was calculated by aligning the union of subsequences from each filter. Considering the biological significances of poly(A) signals, only the positive PAS sequences in the training set were used for PPM calculation. Total 128 PPMs with 12-mer width were retrieved from the first convolutional layer of CNN and subsequently converted to position weight matrix (PWM). PWMs were sorted by entropy values using Desc-Tools and visualized using R package *ggseqlogo*. PWM motif distribution on sequences around PASs was calculated using MEME *centrimo*.

### Comparison of DeepPASS with other models

DeepPASS was compared with other poly(A) site prediction models, such as Deep-PASTA [19] and APARENT [20], to assess its performance in PAS evaluation. The independent validation set with 25,107 stringent PASs (Step 1: training set construction)

that were not used for model training were randomly selected ($n$ = 15,000) for comparison with five repeats. A matched set of 200-bp negative sequences was randomly extracted in intergenic regions (without overlapping with any of the annotated PASs in three databases or GENCODE). For DeepPASTA, input sequences were formatted as previously described [19]. For APARENT, extra 100 Ns were individually added to both the start and the end of each input sequence as APARENT model requires to avoid PASs close to the start or the end of the sequences for scoring (https://github.com/johli/aparent/issues/3). To evaluate their sensitivities and specificities of three models, receiver operating characteristic curves (ROC) were plotted and area under curve (AUC) values were calculated by true positive rates (TPRs) and false positive rates (FPRs) on predicted outcomes.

### Motif analyses of poly(A) signals around called PASs

Four motifs of poly(A) signals, including AAUAAA and its variants (AUUAAA, A[GC]UAAA, AA[GC]AAA), were examined in regions from upstream 50 bp to downstream 25 bp of 3′ ends of called PASs. For their distribution, the 200-bp sequences around 3′ ends of called PASs (− 100 ~ + 100 bp) were extracted and profiled along the sequences using MEME *centrimo* (parameter: --verbosity 1 --norc).

### Quantification of PASs in scRNA-seq dataset

To quantify expression/abundance of PASs, we calculated and normalized the number of mapped reads in its peak region of each PAS. For PASs identified by SCAPTURE or other tools, we directly used the reported peak region of PASs for read counting. For known PASs reported in databases, we collected reads mapped to their upstream 400-bp regions of annotated cleavage sites (usually only one or several nucleotides). If the PAS covers multiple transcripts in a gene, the maximumly expressed region was selected to represent the actual peak region of PAS. Of note, an average 400-bp width of scRNA-seq peaks was observed from multiple 3′ tag-based scRNA-seq datasets [29] and in this study (data not shown). Read counting was conducted by featureCounts (version 1.6.2, parameter: -M -O -F GTF -t exon -g gene_id -s 1 -Q 3) for RPM calculation (normalized by the total number of mapped reads in each sample and multiple by one million).

### Evaluation of PASs identified in scRNA-seq dataset by enrichment scores

Enrichment scores of SCAPTURE-identified PASs were calculated according to a previously published method for PAS quantification from bulk cell RNA-seq [30]. In theory, an accurate PAS is located at the 3′ boundary of its RNA-seq peak, resulting in a high read count in its upstream region and a low read count in its downstream region. In this study, we extended to use this strategy for similar PAS quantification from 3′ tag-based scRNA-seq datasets (Additional file 1: Fig. S7). Read counts within 400-bp upstream or 400-bp downstream region of given PASs were independently obtained as described in the previous section "Quantification of PASs in scRNA-seq dataset," and enrichment scores were calculated as:

$$\text{Enrichment score} = \log 2 \frac{RPM_{upstream} + 0.1}{RPM_{downstream} + 0.1}$$

## Sensitivity and specificity of SCAPTURE evaluated with enrichment scores

To evaluate sensitivity and specificity of the SCAPTURE pipeline, sets of true positive, false positive, true negative, and false negative PASs are required (https://en.wikipedia.org/wiki/Confusion_matrix). Briefly, 34,108 expressed peaks with RPM ≥ 1 called by the first step of SCAPTURE from examined PBMC samples were divided into two groups: "predicted" positive (with probability > 0.5) and "predicted" negative (with probability ≤ 0.5) PAS sites. The same 34,108 peaks were also examined to obtain their enrichment scores as described above. "Actual" positive and negative sites were selected with an enrichment score ≥ 2 or < 2, respectively. These "predicted" and "actual" sites led to a confusion matrix of true positive, false positive, true negative, and false negative for the evaluation of sensitivity and specificity of the DeepPASS model. Different criteria were used for selection of "actual" positive and negative sites with enrichment scores ≥ 1.5, 1.75, 2, 2.25, or 2.5, all resulting in similar sensitivity and specificity of DeepPASS.

## Single-cell analysis with Seurat

All six scRNA-seq PBMC datasets from 10x Genomics were pre-processed by universal tool Cell Ranger [13] (v4.0). Briefly, the genome reference was built using Cell Ranger *mkref* with genome and gene annotation from GENCODE (v34, hg38) with default parameters. Main steps of scRNA-seq data pre-processing, including genome alignment, cell barcode identification and UMIs incorporation, were conducted by using Cell Ranger *count* with default parameters. The generated feature-barcode matrices were used to performed downstream single-cell analysis with the R package Seurat (v3.2.2) [31]. Cell quality control was applied to remove single-cell libraries with low qualities. Three filters were used to perform cell selection: (1) the proportion of mitochondrial counts ≤ 20%; (2) number of UMI counts ≥ 800; (3) number of detected genes ≥ 500. Filtered cells were normalized for subsequent cell clustering performance. Filtered feature-barcode matrices were normalized and then used to calculate gene variation for six PBMC samples individually. Top 2000 of highly variable genes were selected to conduct sample integration using canonical correlation analysis (CCA) by Seurat. After integration, principal component analysis (PCA) was applied to perform linear dimensional reduction and the returned top 50 principal components (PCs) were kept. To apply unsupervised clustering, a KNN graph was built on the top 50 PCs to identify cell clusters with resolution of 0.3 with the *FindClusters* function.

Cell type annotations of different cell clusters in human PBMC samples were based on the expression of marker genes collected from PanglaoDB [45] and/or previous studies [46–48]. Six major cell types, including T cells, B cells, monocytes, natural killer cells, dendritic cells, and platelets, and some of their subtypes were clustered in these human PBMC scRNA-seq datasets from 10x Genomics. To visualize individual cells, uniform manifold approximation and projection (UMAP) was used to project cells in a low-dimensional space. All single cells were labeled with different colors in UMAP plot

according to their samples, cluster identities, and cell type information, respectively (Additional file 1: S9A).

### PAS-based transcript-level single-cell analysis

Since multiple transcripts that harbor distinct PASs can be produced from single-gene loci, values of quantified PASs by SCAPTURE could be theoretically applied to represent expression of corresponding APA transcripts due to different PAS usages. In this case, SCAPTURE could apply DTE values as input to perform single-cell analysis. Several steps, including PAS-based transcript expression matrix calculating, DTE level single-cell analyzing, and comparing with canonical DGE results, were included in the final step of the SCAPTURE pipeline for single-cell clustering.

To generate DTE matrices for six PBMC scRNA-seq datasets from 10x Genomics, single cells that passed quality control step (described in "Canonical single-cell analysis with Seurat") were used to calculate UMI counts per APA transcript in SCAPTURE pipeline. The output transcript expression matrices were loaded into Seurat for downstream analysis. DTE level single-cell analyzing mainly followed the protocol of canonical DGE level analyzing by Seurat. First, for expression quality control, the lowly expressed transcripts with detected cell number ≤ 0.5% were filtered out. The filtered APA transcript matrices were normalized and used to calculate variation of transcripts with distinct PASs. Top 2000 variable APA transcripts were kept and used to perform sample integration with CCA method. Then, the dimensional reduction and unsupervised clustering analysis were applied to single cells based on DTE by SCAPTURE using same parameter with canonical DGE analyzing by Seurat (PCs = 50, resolution = 0.3). Cell clusters by DTE values were subsequently used to assign cell types in PBMCs with collected marker genes.

To evaluate their classification effects on cell type clustering, phylogenetic trees of cell types clustered by DTE or DGE values were individually built by using Seurat *BuildClusterTree* with top 800 variable features. To compare their differences, the cell type-to-barcode information by DTE or DGE values was used to calculate confusion matrix and further visualization (Additional file 9: Table S8). Cells were unassigned by DGE (Fig. 4e–f) were manually sub-grouped to "unassigned to NKCs" and "unassigned to DCs" according to their DTE-based assignment. This re-assignment was further confirmed by their distinct marker gene expression (Additional file 10: Table S9), shown in Fig. 4g, and visualized by an additional phylogenetic analysis (Fig. 4h).

### Proximal PAS usage of single cells

To calculate proximal PAS usage of genes, we quantify expression levels of all identified PASs at single-cell level (Step 3 of "Construction of SCAPTURE pipeline"). These identified PASs were then sorted by their distances to 3′ ends of genes. The proximal PAS usage was mainly defined as the ratio of proximal PAS to the rest of other PASs on a given gene. Altered APA transcript differences between different cell types or conditions were analyzed using Seurat *FindMarkers* function. Genes with an altered proximal PAS usage below certain thresholds (logFC ≥ 1.2, $P$ ≤ 0.05 and min.pct ≥ 0.3) were considered to be significant.

$$\text{Proximal PAS usage} = \log 2 \frac{CPM_{proximal} + 1}{\sum CPM_{rest} + 1}$$

   To globally evaluate the preference of proximal PAS usage between single cells, we calculated the mean value of proximal PAS usage across expressed polyadenylated genes per cell. Then, the preference of proximal PAS of single cells were further scaled by *Z*-score among examined cell groups in scRNA-seq datasets. The resulted score could be further plotted using *UMAP* projection or other statistic methods to represent the degree of proximal PAS usage (3′UTR shortening) of all genes in single cells. In this study, we excluded platelet cells in calculating the preference of proximal PAS usage, due to their low RNA contents (Fig. 5d). As shown in Fig. 5a and e, the monocyte group represented a combination of CD14+ and CD16+ monocytes, the DC group represented a combination of DC, DC2, and plasmacytoid DCs, the T cell group represented a combination of CD4+ and CD8+ T cells, and the B cell group represented a combination of B cells and plasma cells, and the NKC group represented a combination of NKCs and NKT cells.

### Gene ontology analysis

For each immune cell type in PBMCs, we profiled gene sets with differently altered PAS usage between healthy individuals and severe COVID-19 patients. Then, we performed GO analysis of these genes using clusterProfiler *enrichGO* function (ont = "BP," pAdjustMethod = "*fdr*," pvalueCutoff = 0.05, qvalueCutoff = 0.3) and reduced redundancy of GO terms by *simplify* function (cutoff = 0.8, by = "*p.adjust*," select_fun = min). Finally, GO terms with $P \leq 0.05$ in at least one cell type were kept.

### RT-PCR validation of SCAPTURE-identified non-overlapped PASs

Total RNAs from PBMCs of systemic lupus erythematosus (SLE) patients were mixed after Trizol extraction (Life technologies). These patient samples were previously collected from Renji Hospital, Shanghai Jiao Tong University School of Medicine, fulfilling the 1997 American College of Rheumatology (ACR) classification criteria for SLE [49]. The study was approved by the Research Ethics Board of Renji Hospital, Shanghai Jiao Tong University School of Medicine. Written informed consent was signed before sample collection. After treated with DNase I (Ambion, DNA-freeTM kit), total RNAs were reverse-transcribed with SuperScript III (Invitrogen) for cDNAs with a specific RT-primer (GCTGTCAACGATACGCTACGTAACGGCATGACAGTGTTTTTTTTTT TTTTTTTTTT) containing oligo(dT) and adaptor sequences (Additional file 4: Table S3). PCR was performed by using 2 × Taq Plus Master Mix (Vazyme) according to the manufacturer's protocol with gene specific primer sets (Additional file 4: Table S3) for SCAPTURE-identified exonic and intronic PAS validation. Of note, all four intronic PASs and six out of nine of exonic PASs were validated by two rounds of PCR reactions, and correspondingly their negative controls were also amplified by two rounds of PCR reactions (Additional file 4: Table S3). All PCR products were loaded on native Agarose gels (2%) and products with expected sizes were observed for PAS validation.

## Statistical analyses

Statistically significant differences were assessed as described in correspondent figure legends. Normality and unimodality statistic tests in SCAPTURE pipeline were performed using R package diptest (v0.75-7) and nortest (v1.0-4), respectively. All statistic tests were performed with the R platform (v4.0.0).

## Supplementary Information

**Additional file 1:** Figures S1-S10 with figure legends.

**Additional file 2: Table S1.** List of collected known PASs. The known PASs in at least two of three (PolyA_DB3, PolyA-Seq or PolySite 2.0) databases or in the GENCODE (v35) annotation were labeled. And the stringent PASs annotated in all three databases or in the GENCODE (v35) annotation were also labeled. (Related Fig. 2A and S2A).

**Additional file 3: Table S2.** List of called peaks and filtered PASs by SCAPTURE with aggregating six PBMC scRNA-seq datasets from 10x Genomics. The information of PAS filtering by DeepPASS and overlapping with known PASs are labeled. a, aggregated exonic PASs. b, aggregated intronic PASs. (Related Fig. 3a, S4B and S5C).

**Additional file 4: Table S3.** List of experimentally validated PASs. a, list of primers for PCR validation. b, detailed identification of validated PASs for SCAPTURE, Sierra and scAPA. (Related Fig. 3d, S5H and S6D).

**Additional file 5: Table S4.** List of exonic PASs identified by SCAPTURE in six PBMC scRNA-seq datasets from 10x Genomics. SCAPTURE-identified exonic PASs in each PBMC sample were individually listed. (Related Fig. S4A).

**Additional file 6: Table S5.** List of intronic PASs identified by SCAPTURE in six PBMC scRNA-seq datasets from 10x Genomics. SCAPTURE-identified intronic PASs in each PBMC sample were individually listed. (Related Fig. S5B).

**Additional file 7: Table S6.** List of PASs identified by Sierra in six PBMC scRNA-seq datasets from 10x Genomics. Sierra-identified exonic and intronic PASs in each PBMC dataset were individually listed. (Related Fig. 3d and S6C).

**Additional file 8: Table S7.** List of PASs identified by scAPA in six PBMC scRNA-seq datasets from 10x Genomics. scAPA-identified exonic and intronic PASs in each PBMC dataset were individually listed. (Related Fig. 3d and S6C).

**Additional file 9: Table S8.** List of barcode, sample and cell type information of each single cell by DGE and DTE analyses in six PBMC scRNA-seq datasets from 10x Genomics. (Related Fig. 4f and S9A-C).

**Additional file 10: Table S9.** Gene expression of cell markers in integrated six PBMC scRNA-seq datasets from 10x Genomics. a, gene expression of markers of natural killer cells and dendritic cells. b, gene expression of all collected markers of twelve classified and one unassigned cell types (Related Fig. 4 g and S9D).

**Additional file 11: Table S10.** List of barcode, sample and cell type information of each single cell by DGE and DTE analyses in PBMC scRNA-seq datasets from healthy individuals and COVID-19 patients [32]. (Related Fig. 5 and S10).

**Additional file 12: Table S11.** GO terms of altered PAS usage between healthy individuals and COVID-19 patients in each immune cell type of PBMCs (Related Fig. S10B).

**Additional file 13:.** Statistical source data of all figures and supplementary figures.

**Additional file 14:** Review history.

## Acknowledgements
We thank Yang laboratory for discussion.

## Review history
The review history is available as Additional file 14.

## Peer review information

## Authors' contributions
L.Y. conceived and supervised the project; G.-W.L. and F.N. preformed most computational analyses supervised by L.Y and G.-H.Y. constructed deep learning framework supervised by L.Y. and C.-X.L. performed PCR validation supervised by L.-L.C. X. L. participated in COVID-19-related data interpretation. B.T. participated in project design and data interpretation. L.Y. and B.T. wrote the paper with input from G.-W.L. and F.N. All author(s) read and approved the final manuscript.

## Funding
This work was supported by the National Natural Science Foundation of China (NSFC) (31925011, 31730111, 91940306), the Ministry of Science and Technology of China (MoST) (2019YFA0802804) and the Chinese Academy of Sciences (XDB38040300) to L.Y. B.T. was funded by NIH grants (R01 GM084089 and R01 GM129069). L.-L.C. acknowledges the support from the XPLORER PRIZE and the HHMI International Program (55008728). C.-X.L. was funded by China Postdoctoral Science Foundation (CPSF) (Y949603101).

### Availability of data and materials

Methods, including statements of data/code availability and their associated accession codes and references, are available in the "Methods" section file. All scripts used in this project are available at https://github.com/YangLab/SCAPTURE [50] and in Zenodo with DOI: https://doi.org/10.5281/zenodo.5091211 [51], including SCAPTURE pipeline, DeepPASS model, and related codes. Multiple RNA-seq datasets from independent resources were used in this study. A collection of bulk cell RNA-seq, Smart-seq2, and 10x Chromium scRNA-seq datasets of human PBMCs was available from the Gene Expression Omnibus (GEO) with accession number GSE132044 [17]. Another collection of six 10x Chromium scRNA-seq datasets from human PBMCs was downloaded from the 10x Genomics company website at https://support.10xgenomics.com/single-cell-gene-expression/datasets. The third collection of bulk cell RNA-seq datasets of monocytes, B cells, or T cells from PBMCs from SLE patients and normal donors was available from the National Omics Data Encyclopedia with accession number OEP000216 [25]. The final collection related to COVID-19 study was available from the GEO with accession number GSE155673 [32].

## Declarations

### Ethics approval and consent to participate
Not applicable.

### Competing interests
L.Y., G.-W.L., F.N., and G.-H.Y. have filed a patent application (202110251298.3) relating to the published work through Shanghai Institute of Nutrition and Health.

### Author details
[1]CAS Key Laboratory of Computational Biology, Shanghai Institute of Nutrition and Health, University of Chinese Academy of Sciences, Chinese Academy of Sciences, Shanghai 200031, China. [2]State Key Laboratory of Molecular Biology, Shanghai Key Laboratory of Molecular Andrology, CAS Center for Excellence in Molecular Cell Science, Shanghai Institute of Biochemistry and Cell Biology, University of Chinese Academy of Sciences, Chinese Academy of Sciences, 320 Yueyang Road, Shanghai 200031, China. [3]Institute of Pathology and Southwest Cancer Center, Southwest Hospital, Third Military Medical University (Army Medical University), Chongqing 400038, China. [4]School of Life Science and Technology, ShanghaiTech University, Shanghai 201210, China. [5]School of Life Science, Hangzhou Institute for Advanced Study, University of Chinese Academy of Sciences, Hangzhou 310024, China. [6]Program in Gene Expression and Regulation, and Center for Systems and Computational Biology, The Wistar Institute, Philadelphia, PA 19104, USA.

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

## 
