## [**Additional file 14:** Review history. · Genome Biology]

Review History

First round of review

Reviewer #1: The authors present a deep learning based method to identify polyadenylation sites (PAS) in 3'-end enriched single cell RNA-sequencing data. The method starts with peak calling to capture regions likely close to PAS. The peaks are then passed through a deep learning strategy to select high confidence PASs. The authors used the quantification of PAS-associated transcripts to cluster single cells, demonstrating that transcript-level quantification could provide additional information not captured by gene level expression analysis. New methods to identify PASs in scRNA-seq data definitely serve a need of the field, and incorporating deep learning methods to enhance the performance in PAS ID is a good idea. However, there are some major concerns that need to be addressed.

The performance of SCAPTURE on scRNA-seq data needs to be evaluated in a much more rigorous manner. The authors mainly use PAS motifs and sequence enrichment to support their performance. However, the deep learning scheme in SCAPTURE uses sequences around PASs to train the model. Motifs around PASs are expected to be main features selected by deep learning to make the predictions of PAS. Thus, using the motifs around the predicted PASs to justify their method is circular. The method should be evaluated using ways that are independent of motifs or nucleotide biases, preferably using data with experimentally supported PASs.

A claimed advantage of SCAPTURE is its de novo nature. However, it is not clear how well it performs in predicting novel PASs in scRNA-seq. What are the sensitivity and specificity here? The only place where novel PASs were shown is Fig. 3, but the evaluation in Figure 3B for non-overlapped PASs is not convincing, nor a fair way of evaluating performance.

Related to the above point, the large number of intronic PASs identified by SCAPTURE is concerning, where 87.8% do not overlap with known intronic PASs. The enrichment of AAUAAA and its variants is not adequate to support the validity of these PASs. More rigorous evaluation is needed. Similarly, non-motif based evaluation is needed in comparing SCAPTURE to Sierra and scAPA.

How precise can SCAPTURE predict the exact location of PASs? In the method, the authors stated that a predicted PAS is considered overlapping known sites if a known site is between "upstream 50 bp to downstream 25 bp" of the predicted PAS. This range seems to be quite large. Is the same range applied in the other methods used for comparison? How effectively can SCAPTURE resolve alternative PASs that are close to each other?

In applying SCAPTURE to cluster cells, the authors showed that PAS-based transcript quantification helped to cluster a small number of cell that were not clustered by gene-level quantification. The result seems to have only incremental improvement. Does transcript based clustering offer any major advantages, such as identifying rare cell types, enhancing the

robustness of the clustering, or requiring less read coverage while maintaining similar sensitivity in cell type segregation?

In using DTEs to cluster cells, were distinct PASs of the same gene in different cells merged together if they are within certain distance? It would be good to provide more details on how this was performed.

Reviewer #2: Alternative polyadenylation (APA) is widespread in eukaryotic mRNA genes. Recognized as major gene regulation, APA plays an important role in a variety of physiological and pathological conditions. In the manuscript by Li et al., the authors reported a stepwise deep learning-embedded pipeline to capture polyadenylation called SCAPTURE from 3'tag-based RNA-seq in the context of single cells. SCAPTURE offers the ability to detect not only annotated polyadenylation sites (PASs) but also novel PASs, thus enabling discovery of APA events under different biological conditions. Additionally, it helps to quantify alternative PAS transcripts which can be used to reflect a knowledge base of cell identities besides commonly used gene expressions from single-cell RNA-seq. This work has the potential to be a valuable tool to study the regulation of APA in single cells. However, several issues need to be addressed before it can be accepted for publication in *Genome Biology*.

Major items:

1. When applying SCAPTURE to profile PASs from publicly available scRNA-seq datasets of human PMBCs from 10x Genomics, the authors stated that higher fractions of both overlapped and non-overlapped exonic PASs identified by SCAPTURE harbored canonical PAS motif of AAUAAA or its variants at much higher level than that of SCAPTURE-rejected exonic sites. The authors suggest this as an indication the SCAPTURE-identified non-overlapped PASs might be bona fide PASs. They also used similar arguments for the intronic PASs and for the evaluation of different PAS analysis methods. My understanding is that the authors are using motif preference and nucleotide distribution of known PAS as a proxy for PAS detection accuracy.

However, since the pipeline was trained by "sequences shifting around known PASs with stringent filtering" which are known to be enriched for the AAUAAA motif, thus, here they run the risk of circular argument that using features enriched for AAUAAA will more likely result in identifying sites enriched for AAUAAA. So even the languages the authors used were tentative/speculative. I would feel more convinced if the authors could provide experimental validations to validate a few such non-overlapping (novel) exonic and intronic PASs to strengthen these statements.

2. The authors proposed the application of SCAPTURE to be facilitating single cell clustering, which did show nice performance in reducing unassigned cells and result in a cleaner clustering result. However, this application by itself does not seem to be substantial enough to showcase the method's utility. I would like to see a little more in-depth investigation, for example, similar to Feng et al (<https://www.pnas.org/content/118/10/e2013056118/>) to construct the regulatory network of APA events in single cells and investigate the mechanism that generated such post-transcriptional APA heterogeneity in complex cell populations.

Minor items:

1. In the training dataset construction, what is the ratio of the negative training dataset over the total training dataset? The imbalanced training data may potentially cause the inflation of the performance of the model. If it is an imbalanced data, it would be helpful to add another metric, AUPRC, which is usually useful for evaluating the performance of imbalanced data.
2. The selection of the negative training dataset may prevent the model from learning more complicated sequence contexts of PASs. As the negative sequences are sampled only from the intergenic region, they are not reflecting the actual sequence compositions in CDS or UTR regions. It would be beneficial to select negative sequences near the PASs sites or at least include some of them in the negative training dataset.
3. The difference between DeepPASS and DeepPASS-fixed is constructing positive training dataset. It seems that the positive sequences in DeepPASS training dataset have a broader region from the center of PASs than the positive sequences in DeepPASS-fixed. Would it be possible to expand the input sequence window (for example, from 200bp to 400bp) in DeepPASS-fixed and see whether the performance is improved or not?
4. In "Comparison of DeepPASS with other models" section, do the 50,000 sequences contain the sequences presented in the training dataset of DeepPASS? Also, what is the standard error if the analysis is randomly repeated a few times?
5. In step 2 "PAS evaluating with an embedded deep learning neural network", please specify the threshold of the DeepPASS prediction for calling high-confidence PASs?
6. Figure 1C legend does not match figure (top/middle/bottom◊ left/middle/right
7. Please specify with more details in the method section what was done during "manual subgrouping" to re-cluster the "unassigned cells"

Authors Response

We like to thank both reviewers for their positive comments and insightful suggestions, which have guided us to improve this manuscript. In this revised manuscript, we have addressed all the points raised by reviewers, with the addition of new experiments and analyses.

Two major and common concerns were raised by both reviewers. One is to provide additional validation, independent of motifs or nucleotide bases, to evaluate the performance of SCAPTURE. The other is to further show advantages/utilities of SCAPTURE for scRNA-seq analysis. We have now performed suggested experiments and analyses to address these comments. Briefly, we have evaluated SCAPTURE more rigorously as suggested, including providing new lines of experimental evidence, which are independent of motifs or nucleotide sequences, to confirm novel exonic and intronic PASs identified by SCAPTURE (Rebuttal_Fig. 1-3; new Fig. 3c-3d and Additional file 1: Fig. S5F-5H and S6B-6D). In addition, we have also performed in-depth analysis to show the advantage of using SCAPTURE to identify altered PAS events in different conditions, such as cases in PBMCs between healthy individuals and COVID-19 patients (Rebuttal_Fig. 10; revised Fig. 5 and Additional file 1: Fig. S10). This highlights the potential of using SCAPTURE to identify alternative PASs as an additional molecular signature of gene expression changes. Moreover, we have performed all other suggested analyses to further evaluate SCAPTURE and the DeepPASS model (Rebuttal_Fig. 5-9 and 11-13). Finally, all the key suggested results have now been included in the revised manuscript. We think this newly developed SCAPTURE pipeline will be of broad utility in analyzing scRNA-seq and revealing underestimated PASs.

Please find our point-by-point responses below. We have also made corresponding changes in the revised manuscript with tracks.

Reviewer #1: The authors present a deep learning based method to identify polyadenylation sites (PAS) in 3'-end enriched single cell RNA-sequencing data. The method starts with peak calling to capture regions likely close to PAS. The peaks are then passed through a deep learning strategy to select high confidence PASs. The authors used the quantification of PAS-associated transcripts to cluster single cells, demonstrating that transcript-level quantification could provide additional information not captured by gene level expression analysis. New methods to identify PASs in scRNA-seq data definitely serve a need of the field, and incorporating deep learning methods to enhance the performance in PAS ID is a good idea. However, there are some major concerns that need to be addressed.

We thank reviewer #1 for his/her general support, and have now addressed all raised concerns with addition of new experiments and analyses.

1. The performance of SCAPTURE on scRNA-seq data needs to be evaluated in a much more rigorous manner. The authors mainly use PAS motifs and sequence enrichment to support their performance. However, the deep learning scheme in SCAPTURE uses sequences around PASs to train the model. Motifs around PASs are expected to be main features selected by deep learning to make the predictions of PAS. Thus, using the motifs around the predicted PASs to justify their method is circular. The method should be evaluated using ways that are independent of motifs or nucleotide biases, preferably using data with experimentally supported PASs.

Thanks for this constructive and helpful suggestion to evaluate the SCAPTURE pipeline with experimentally supported PASs. In this revised manuscript, we have now provided experimental evidence supporting the existence of non-overlapped/novel exonic and intronic PASs identified by SCAPTURE by PCR amplification of PBMC cDNA samples from SLE patients (originally reported by Liu et al., *Cell* 2019). These experiments were performed by Dr. Chu-Xiao Liu at Dr. Ling-Ling Chen's lab, who are also listed as co-authors in the revised manuscript.

Briefly, four SCAPTURE-identified novel exonic PASs were manually selected for experimental validation in the revised manuscript, which passed the selection cutoff of the DeepPASS model (with a predicted probability of 0.85~0.99) and showed high peaks called by SCAPTURE (Rebuttal_Fig. 1A). Mixed PBMC RNA samples from several SLE patients previously examined by Liu et al (*Cell* 2019) were used for this RT-PCR validation with specific primers (Rebuttal_Fig. 1B), according to a previously-reported strategy in designing corresponding primers (Masamhali et al., *Nature* 2014). As shown in Rebuttal_Fig. 1C, all these SCAPTURE-identified novel exonic PASs could be successfully confirmed by corresponding PCR products with correct sizes. As a positive control, a reported (proximal) PAS in the last exon of *NADK* gene was also confirmed by

Rebuttal_Fig. 1. Experimental validation of SCAPTURE-identified non-overlapped/novel exonic PASs. (A) Comparison of different transcriptome profiling around four SCAPTURE-identified novel exonic PASs. PCR primers are shown in the bottom for experimental validation. Asterisks, non-specific RNA-seq signals due to multiple mapping. (B) Schematic diagram of primer design for PCR validation. (C) PCR results to validate SCAPTURE-identified novel exonic PASs. Of note, additional nested primers were designed to validate the SCAPTURE-identified PAS in the *PPP2R5C* locus.

this PCR method (Rebuttal_Fig. 1C). Meanwhile, as negative controls, PCR signals from downstream regions of SCAPTURE-identified novel PASs were undetectable (Rebuttal_Fig. 1C).

Of note, these exonic PASs examined in Rebuttal_Fig. 1 could be also called by other tools Sierra (four out of four) or scAPA (two out of four). To further confirm the reliability of SCAPTURE, five additional novel exonic PASs, which were uniquely identified by SCAPTURE, but not by

Sierra or scAPA, were manually selected (with a predicted probability of 0.58~0.88 and scRNA-seq peaks called by SCAPTURE) for validation by PCR (Rebuttal_Fig. 2A). Due to their low abundance, nested PCR primers were designed for the validation (Rebuttal_Fig. 2B). As shown in Rebuttal_Fig. 2C, all these novel exonic PASs uniquely-identified by SCAPTURE could be successfully confirmed by corresponding PCR products with correct sizes (Rebuttal_Fig. 2C). Meanwhile, as negative controls, no PCR products that were examined by the same nested PCR reaction at downstream regions of these five exonic PASs were detected (Rebuttal_Fig. 2C).

Similar PCR validation was also applied to four additional SCAPTURE-identified novel intronic PASs, which passed the cutoff of the DeepPASS model with a predicted probability of 0.58~0.92 and showed scRNA-seq peaks called by SCAPTURE (Rebuttal_Fig. 3A). Due to their low abundance,

Rebuttal_Fig. 2. Experimental validation to show non-overlapped/novel exonic PASs uniquely identified by SCAPTURE. (A) Comparison of different transcriptome profiling around five SCAPTURE-unique novel exonic PASs. PCR primers are shown in the bottom for experimental validation. (B) Schematic diagram of primer design for PCR validation. (C) PCR results to validate SCAPTURE-unique novel exonic PASs.

Rebuttal_Fig. 3. Experimental validation to show SCAPTURE-identified non-overlapped/novel intronic PASs. (A) Comparison of different transcriptome profiling around four SCAPTURE-identified novel intronic PASs. PCR primers are shown in the bottom for experimental validation. (B) Schematic diagram of primer design for PCR validation. (C) PCR results to validate SCAPTURE-identified novel intronic PASs. Asterisks, non-specific PCR bands.

nested PCR primers were designed to validate SCAPTURE-identified novel intronic PASs (Rebuttal_Fig. 3B). As shown in Rebuttal_Fig. 3C, all these SCAPTURE-identified novel intronic PASs could be successfully confirmed by corresponding PCR products with correct sizes (Rebuttal_Fig. 3C). As negative controls, PCR signals from downstream regions of SCAPTURE-identified novel intronic PASs were barely detectable.

Other than PCR validation (Rebuttal_Fig. 1-3), we have also collected several scRNA-seq datasets from human tissues (Liao et al., *Sci Data* 2020; MacParland et al., *Nat Commun* 2018; Sohni et al., *Cell Rep* 2019) for additional validation. Preliminary results showed that seven out of nine validated novel exonic PASs in rebuttal Fig. 1 and 2 could be also clearly detected in human kidney, liver and testis scRNA-seq samples (Rebuttal_Fig. 4A and 4B), while two out of four validated novel intronic PASs in rebuttal Fig. 3 could not be detected in human scRNA-seq samples (Rebuttal_Fig. 4C), due possibly to their tissue-specific expression.

Taken together, we have now provided substantial data to experimentally support the existence of SCAPTURE-identified, novel PASs, which are completely independent of motifs or nucleotide bases. We have now included these validation results in the revised Fig. 3c-3d; Additional file 1: Fig. S5F-5H and Fig. S6B-6D. Due to space limitation, the preliminary finding of PAS peaks corresponding to these novel PASs in human kidney, liver and testis scRNA-seq samples (Rebuttal_Fig. 4) are not included in the revised manuscript. However, if the reviewer and editor think it is necessary, we would be happy to add them in.

Rebuttal_Fig. 4. Detection of experimentally validated novel exonic PASs (A), novel exonic PASs that uniquely identified by SCAPTURE (B), and novel intronic PASs (C) in human tissue scRNA-seq datasets.

2. A claimed advantage of SCAPTURE is its de novo nature. However, it is not clear how well it performs in predicting novel PASs in scRNA-seq. What are the sensitivity and specificity here? The only place where novel PASs were shown is Fig. 3, but the

evaluation in Figure 3B for non-overlapped PASs is not convincing, nor a fair way of evaluating performance.

Thanks for the question. As suggested, we have now provided experimental evidence to confirm the existence of several SCAPTURE-identified novel PASs (Please also see our responses to the above comment, Rebuttal_Fig. 1-4). Other than this experimental validation, we have performed additional analyses to comprehensively evaluate the SCAPTURE method in this revised manuscript, including its sensitivity and specificity.

One of characteristic features of SCAPTURE is incorporation of a deep learning neural network (DeepPASS) to evaluate identified PASs. Instead of using all (251,071) of stringent PASs in the previous version for DeepPASS training (Rebuttal_Fig. 5A, left), in the revised manuscript, stringent PASs were split with a 9:1 ratio between the model training and independent validation sets (225,964 and 25,107, respectively) (Rebuttal_Fig. 5A, right). Similar results were obtained when using all or 90% of stringent PASs for this model training. For example, compared with commonly-used methods that are based on fixed sequences for feature extraction (termed DeepPAS-fixed here), DeepPASS trained with all or 90% of stringent PASs both achieved higher area under curve (AUC)

values (Rebuttal_Fig. 5B). Meanwhile, when using 10% of stringent PASs that were not used for the model training for an independent comparison, AUC values of DeepPASS trained with all or 90% of stringent PASs were substantially higher than two other previously-reported methods, such as DeepPASTA and APARENT (Rebuttal_Fig. 5C).

To evaluate the sensitivity and specificity of the DeepPASS model, sets of true positive, false positive, true negative and false

Rebuttal Fig. 5. Similar performances by using all (A) or 90% (B) of stringent PASs for DeepPASS model training. (C) Five times of independent validation were performed to show consistent AUC values with very low standard errors for all three models.

negative PASs are needed (https://en.wikipedia.org/wiki/Confusion_matrix). We focused on 34,108 highly-expressed PAS peaks with $RPM \geq 1$ (out of 83,390 with $RPM > 0$) that were called by the first step of SCAPTURE from PBMC samples to construct this confusion matrix (Rebuttal_Fig. 6A). On the one hand, the 34,108 expressed peaks that passed the DeepPASS model were divided into two groups: DeepPASS-predicted positive and DeepPASS-predicted negative PAS sites, according to DeepPASS-predicted probability scores. Briefly, a predicted probability ranging between 0 and 1 of a given candidate PAS was obtained with the DeepPASS model, which was further classified into a group of positive sites with predicted probability > 0.5 or a group of negative sites with predicted probability ≤ 0.5 (Please also see the revised “methods” section). On the other hand, the 34,108 expressed peaks were also examined to obtain their enrichment scores (Rebuttal_Fig. 6B), according to a previously-published method to quantify and evaluate PASs (Irtisha Singh et al., *Nat Commun* 2018). Similar to this previous study by Irtisha Singh et al (*Nat Commun* 2018), we have used enrichment score as cutoff to obtain “actual” positive (enrichment score ≥ 2) and “actual” negative (enrichment score < 2) PASs (Rebuttal_Fig. 6A). Based on this criterion, we obtained a confusion matrix of true positive, false positive, true negative and false negative for evaluation of sensitivity and specificity of the DeepPASS model. According to the formulae listed in the bottom of Rebuttal_Fig. 6A, the DeepPASS model could achieve a sensitivity score of 0.69, a specificity score of 0.66, and an F_1 score of 0.74 (Rebuttal_Fig. 6C). Similar sensitivity and specificity scores were also obtained when using human kidney, liver and testis scRNA-seq datasets for analyses (Rebuttal_Fig. 6C). Of note, very similar sensitivity and specificity values were obtained with the DeepPASS model when using different enrichment scores (≥ 1.5 ,

Rebuttal_Fig. 6. Calculation of DeepPASS's sensitivity and specificity. (A) Pipeline of using enrichment scores to evaluate sensitivity and specificity of DeepPASS. (B) Schematic diagram of enrichment score calculation. (C) Sensitivity, specificity and F_1 scores of DeepPASS calculated individually from PBMC, kidney, liver and testis scRNA-seq datasets.

1.75, 2, 2.25 or 2.5, respectively) as cutoffs for “actual” positive/negative PAS sites (Rebuttal_Fig. 6C).

To further compare SCAPTURE with Sierra and scAPA, we also calculated the ratio of each tool for recapturing known PASs in database. In training and validation sets (251,071 of stringent PASs split with a 9:1 ratio), ~71% and 70% of expressed PASs in examined PBMCs with RPM ≥ 1 (calculated from Cell Ranger mapped BAM files within upstream 400 bp of stringent PASs) could be respectively identified by SCAPTURE, indicating that the sensitivity of SCAPTURE (Rebuttal_Fig. 7) is similar to the value calculated by the enrichment score in Rebuttal_Fig. 6C. In contrast, the sensitivity scores of Sierra and scAPA were only about 60% and 47%, respectively (Rebuttal_Fig. 7). Of note, we also have provided experimental evidence to confirm five predicted PASs uniquely-identified by SCAPTURE, but not by Sierra or scAPA (Rebuttal_Fig. 2 and 4B). These results together indicate that SCAPTURE has a higher sensitivity in PASs calling.

Rebuttal_Fig. 7. SCAPTURE recaptured relatively high numbers of stringent PASs than other tools. (A) Statistics of identified PASs from training and validation datasets (split with a 9:1 ratio from stringent PASs) by SCAPTURE, Sierra and scAPA. (B) Bar plots of identified PAS rates by SCAPTURE, Sierra and scAPA (statistics from A).

3. Related to the above point, the large number of intronic PASs identified by SCAPTURE is concerning, where 87.8% do not overlap with known intronic PASs. The enrichment of AAUAAA and its variants is not adequate to support the validity of these PASs. More rigorous evaluation is needed. Similarly, non-motif based evaluation is needed in comparing SCAPTURE to Sierra and scAPA.

Thanks for this comment. It is well known that many false positive intronic PAS sites would be called in the intronic regions due to internal priming at internal A-rich sequences, despite specific methods to remove such PASs (Nam et al, *Proc Natl Acad Sci U S A* 2002; Patrick et al, *Genome Biol* 2020). We have applied stringent cutoffs in the SCAPTURE pipeline to remove these false positive sites, including those containing A-rich context sequences (consecutive adenines ≥ 8). Compared to Sierra and scAPA, SCAPTURE identified significantly fewer intronic PASs (16,082 vs 111,039/35,884).

Rebuttal_Fig. 8. Overlapping of intronic PASs (blue) identified from six PBMC datasets from 10x Genomics by different computational tools with known intronic PASs in databases (purple). SCAPTURE (left), Sierra (middle) and scAPA (right).

Among them, $\sim 18.6\%$ of SCAPTURE-identified intronic PASs overlapped with known sites, compared to $\sim 2.1\%$ or 2.0% for Sierra- or scAPA- identified intronic PASs (Rebuttal_Fig. 8). In view of the technical difficulties in calling intronic PASs, we now mainly discuss intronic PAS identification by SCAPTURE in the additional, but not main, figures.

Nevertheless, in this revised manuscript, we have also provided experimental validation to confirm the existence of four SCAPTURE-identified novel intronic PASs (Rebuttal_Fig. 3 and 4C). Of note, no consecutive adenines (>6) were identified in the context regions of all these four examined SCAPTURE-identified novel intronic PASs (Rebuttal_Fig. 3 and 4C), suggesting that they were not likely to be false positives caused by internal priming at A-rich sequences.

Despite our new effort, we think that additional attention should be paid in the study of intronic PASs, perhaps using specifically developed methodologies.

4. How precise can SCAPTURE predict the exact location of PASs? In the method, the authors stated that a predicted PAS is considered overlapping known sites if a known site is between "upstream 50 bp to downstream 25 bp" of the predicted PAS. This range seems to be quite large. Is the same range applied in the other methods used for comparison? How effectively can SCAPTURE resolve alternative PASs that are close to each other?

Thanks for the question. It has been reported that nucleotides around cleavage sites have distinct patterns (Gruber and Zavolan, *Nat Rev Genet* 2019), which is indicated in Rebuttal_Fig. 9A. In our hands, both SCAPTURE and Sierra could accurately identify the exact location of PASs (Rebuttal_Fig. 9B, and revised Fig. 3B and 3F). It is also well known that the polyadenylation signal is usually located ~ 21 bp upstream of the cleavage site (Wang et al, *Genome Res* 2018). Profiling the canonical polyadenylation signal (AAUAAA) and its variants showed an

Rebuttal_Fig. 9. SCAPTURE could accurately identify the location of PASs. (A) Reported nucleotide pattern around cleavage sites. (B) Nucleotide around exonic PASs identified by different computational tools. (C) Density of polyadenylation signal and its variants around PASs by different computational tools. Relative positions to cleavage sites were labeled in plot. (D) Distances of adjacent alternative PASs. Number of alternatively polyadenylated genes were labeled.

enrichment of polyadenylation signals located in the ~50 bp (with a peak at ~18-19 bp) nucleotide region upstream to predicted PASs by SCAPTURE and Sierra (Rebuttal_Fig. 9C), further suggesting the accuracy for PAS calling. In contrast, the enrichment was not clearly observed by using scAPA for prediction. In summary, we concluded that both SCAPTURE and Sierra could identify the exact location of PASs.

Of note, here we screened "upstream 50 bp to downstream 25 bp" regions of tool-predicted PASs to identify known PASs. If there is a known PAS within "upstream 50 bp to downstream 25 bp" of a predicted PAS, it would be labeled as an "overlapped site". We applied this setting (a relatively wide window) to avoid mislabeling overlapped sites as non-overlapped sites. The same window setting was used in comparison of all three tools.

As suggested, we further compared how SCAPTURE, Sierra and scAPA performed to resolve alternative PASs. We counted distances between two adjacent PASs and found that most of PASs (96.6%, 87.7% and 99.9% for SCAPTURE, Sierra or scAPA, respectively) were > 100 bp away by using all three tools based on thousands of gene loci (Rebuttal_Fig. 9D). This suggested that all three tools, including newly-developed SCAPTURE, could identify alternative PASs with at least 100-bp distance.

5. In applying SCAPTURE to cluster cells, the authors showed that PAS-based transcript quantification helped to cluster a small number of cell that were not clustered by gene-level quantification. The result seems to have only incremental improvement. Does transcript based clustering offer any major advantages, such as identifying rare cell types, enhancing the robustness of the clustering, or requiring less read coverage while maintaining similar sensitivity in cell type segregation?

Thanks a lot for the question. We agree that it is a relatively minor improvement to use PAS-based transcript quantification as compared to gene-level quantification in our previous submission. Since SCAPTURE is designed to identify differential transcript expression (DTE) rather than differential gene expression (DGE) with canonical scRNA-seq analysis tools, we think that the major advantage of using SCAPTURE is to identify altered PAS usage among conditions. For example, by using the same benchmark scRNA-seq datasets of normal PBMCs from 10x Genomics, we found that monocytes have a less proximal PAS usage than other major types of immune cell types, such as dendritic cells (DCs), in PBMCs (Rebuttal_Fig. 10A).

In this revised manuscript, we have further applied SCAPTURE to analyze published scRNA-seq datasets of PBMCs from healthy individuals and COVID-19 patients (Arunachalam et al., *Science* 2020) to show its advantage in identifying APA events

(Rebuttal_Fig. 10). With scRNA-seq datasets of PBMCs from Arunachalam et al (Arunachalam et al., *Science* 2020), a similar finding of less proximal PAS usage in monocytes was also observed in PBMCs independently from healthy individuals or COVID-19 patients (Rebuttal_Fig. 10B and 10C). The different proximal PAS usages between monocytes and DCs at some known gene loci with reported functions in activation of innate/adaptive immune response, antigen processing/presentation via MHC or dendritic spine development were highlighted in Rebuttal_Fig. 10D, suggesting their distinct roles among different types of immune cells.

Rebuttal_Fig. 10. Application of SCAPTURE showed altered PAS usage between healthy and COVID-19 patients. Proximal PAS usage of (A) six scRNA-seq datasets of normal PBMCs from 10x Genomics, (B) scRNA-seq datasets of PBMCs from healthy donors and (C) scRNA-seq datasets of PBMCs from COVID-19 patients. (D) altered PAS usage between monocytes and dendritic cells. (E) Preferential usage of proximal PASs in immunoglobulin genes with SARS-CoV-2 infection.

Strikingly, compared to those in healthy individuals, we also observed a general preference of using proximal PASs, suggesting 3'UTR shortening, in all major PBMC cell types of COVID-19 patients (Rebuttal_Fig. 10E). This result was in line with previous bulk cell data indicating preferential activation of proximal PASs in PBMCs upon infections (Jia et al., *Nat Commun* 2017; Pai et al., *Plos Genet* 2016). Interestingly,

a subgroup of genes, including *IGHM*, *IGHG1*, *IGHG3* and *IGHA2*, show distinct PAS usages in PBMCs between healthy individuals and COVID-19 patients, specifically in plasma cells (Rebuttal_Fig. 10F). Of note, it has been previously reported that the usage of proximal PAS in immunoglobulin heavy chain genes, such as *IGHM* (Early et al., *Cell* 1980; Rogers et al., *Cell* 1980), is important to their functions. For example, the *IGHM* transcript with its proximal PAS, which replaces two terminal exons for transmembrane domain with a hydrophilic secretory tail, leads to the translation of a secretion form of IgM antibody involved in neutralization. Strikingly, some recent studies reported the role of secreted IgM and IgA antibodies in SARS-CoV-2 neutralization (Garcia-Beltran et al., *Cell* 2021; Srerlin et al., *Sci Transl Med* 2021; Wang et al., *bioRxiv* 2020). Thus, our finding of the preferential proximal PAS usage at these immunoglobulin heavy chain gene loci is in line with reported immune response of these genes upon SARS-CoV-2 infection. Prof. Xindong Liu, who is an expert in the immunology field, was involved in this part of data analysis, and thus has been added as a co-author in the revised manuscript.

Collectively, APA/PAS-transcript changes observed by SCAPTURE thus highlighted additional molecular signatures of gene expression changes at the PAS level. While we present several important biological messages in this revised manuscript, we do wish to emphasize that the main goal of work is to present an efficient pipeline to identify PASs from scRNA-seq datasets and its application in identifying distinct PAS usages. We hope the reviewer is satisfied with our in-depth analysis in Rebuttal_Fig. 10 (revised Fig. 5 and Additional file 1: Fig. S10) and is in favor of publication of our work as a Method type paper at Genome Biology.

6. In using DTEs to cluster cells, were distinct PASs of the same gene in different cells merged together if they are within certain distance? It would be good to provide more details on how this was performed.

Thanks for raising this question and we apologize for not having clearly described detailed methods in the previous submission. SCAPTURE, like many other peak calling-based tools, uses Cell Ranger mapped BAM file as input to represent combined signals from all cells in a sample. In this case, distinct PASs of the same gene in different single cells were merged together if they had an overlap ratio $\geq 50\%$, and only the peak with highest expression levels was used for downstream analysis. We have now included this detail in the revised manuscript (revised “method” section with track).

Reviewer #2: Alternative polyadenylation (APA) is widespread in eukaryotic mRNA genes. Recognized as major gene regulation, APA plays an important role in a variety of physiological and pathological conditions. In the manuscript by Li et al., the authors

reported a stepwise deep learning-embedded pipeline to capture polyadenylation called SCAPTURE from 3'tag-based RNA-seq in the context of single cells. SCAPTURE offers the ability to detect not only annotated polyadenylation sites (PASs) but also novel PASs, thus enabling discovery of APA events under different biological conditions. Additionally, it helps to quantify alternative PAS transcripts which can be used to reflect a knowledge base of cell identities besides commonly used gene expressions from single-cell RNA-seq. This work has the potential to be a valuable tool to study the regulation of APA in single cells. However, several issues need to be addressed before it can be accepted for publication in Genome Biology.

We thank reviewer #2 for his/her positive comments and general support. We have now addressed all raised concerns with the addition of new experiments and analyses.

Major items:

When applying SCAPTURE to profile PASs from publicly available scRNA-seq datasets of human PMBCs from 10x Genomics, the authors stated that higher fractions of both overlapped and non-overlapped exonic PASs identified by SCAPTURE harbored canonical PAS motif of AAUAAA or its variants at much higher level than that of SCAPTURE-rejected exonic sites. The authors suggest this as an indication the SCAPTURE-identified non-overlapped PASs might be bona fide PASs. They also used similar arguments for the intronic PASs and for the evaluation of different PAS analysis methods. My understanding is that the authors are using motif preference and nucleotide distribution of known PAS as a proxy for PAS detection accuracy.

1. However, since the pipeline was trained by "sequences shifting around known PASs with stringent filtering" which are known to be enriched for the AAUAAA motif, thus, here they run the risk of circular argument that using features enriched for AAUAAA will more likely result in identifying sites enriched for AAUAAA. So even the languages the authors used were tentative/speculative. I would feel more convinced if the authors could provide experimental validations to validate a few such non-overlapping (novel) exonic and intronic PASs to strengthen these statements.

Thanks for raising this point, which is also raised by reviewer #1. In this revised manuscript, we have now provided experimental evidence to validate a few non-overlapped (novel) exonic and intronic PASs identified by SCAPTURE using PCR and PBMC cDNA samples from human SLE patients (Rebuttal_Fig. 1-3). Briefly, nine SCAPTURE-identified novel exonic PASs (Rebuttal_Fig. 1 and 2) and four additional SCAPTURE-identified novel intronic PASs (Rebuttal_Fig. 3) could be successfully confirmed by corresponding PCR products with correct sizes (Rebuttal_Fig. 1C, 2C and 3C, respectively). Of note, some of these novel PASs sites were uniquely identified by SCAPTURE only (Rebuttal_Fig. 2). Please also see our responses to point #1 of

reviewer #1. We hope the addition of this experimental validation that is independent of motifs could further strengthen our statements.

2. The authors proposed the application of SCAPTURE to be facilitating single cell clustering, which did show nice performance in reducing unassigned cells and result in a cleaner clustering result. However, this application by itself does not seem to be substantial enough to showcase the method's utility. I would like to see a little more in-depth investigation, for example, similar to Feng et al (<https://www.pnas.org/content/118/10/e2013056118/>) to construct the regulatory network of APA events in single cells and investigate the mechanism that generated such post-transcriptional APA heterogeneity in complex cell populations.

Thanks for this point and very constructive suggestion about further investigation of our method's utility. We appreciate the reference of Feng et al., which represents an exemplary study in showing how alternative RNA processing, in addition to DEG, can be used as molecular signatures to define neuronal identity and functions. We truly enjoyed reading this elegant work. In our revised manuscript, we have applied SCAPTURE for some additional analyses. Since SCAPTURE is designed to identify DTE, we think that the major advantage of using SCAPTURE is to identify altered PAS usage among conditions. We have now shown the general proximal PAS usages in monocytes from both healthy individuals and COVID-19 patients (Rebuttal_Fig. 10A-10C). Strikingly, compared to those in healthy individuals, we also observed a general preference of using proximal PASs, suggesting 3'UTR shortening, in all major PBMC cell types of SARS-CoV-2-infected individuals (Rebuttal_Fig. 10E). This result was in line with previous bulk cell data indicating preferential activation of proximal PASs in PBMCs upon infections (Jia et al., *Nat Commun* 2017; Pai et al., *Plos Genet* 2016). A subgroup of genes, including *IGHM*, *IGHG1*, *IGHG3* and *IGHA2*, show distinct PAS usages in PBMCs between healthy individuals and COVID-19 patients, specifically in plasma cells (Rebuttal_Fig. 10F). Our finding of the preferential proximal PAS usage at these immunoglobulin heavy chain genes loci suggests a similar immune response of these genes upon SARS-CoV-2 infection. Please also see our responses to point #5 of reviewer #1. We hope the reviewer is also satisfied with our in-depth analysis in Rebuttal_Fig. 10 and revised Fig. 5 and is in favor of publication of our work as a Method type paper at Genome Biology.

Minor items:

1. In the training dataset construction, what is the ratio of the negative training dataset over the total training dataset? The imbalanced training data may potentially cause the inflation of the performance of the model. If it is an imbalanced data, it would be helpful

to add another metric, AUPRC, which is usually useful for evaluating the performance of imbalanced data.

Thanks for the question and we apologize for not having clearly described this in previous submission. In both previous and revised versions, the ratio of positive training set to negative training set is set to $\sim 1:1$, which is balanced. In the revised manuscript, 2,259,640 of 200-bp positive sequences (~ 10 sequences shifting around each of 225,964 stringent PASs) and exactly the same amount of 2,259,640 negative sequences were used for the DeepPASS model training. Meanwhile, 225,964 positive sequences with fixed position around these 225,964 stringent PASs (-100 \sim +100 bp) and exactly the same amount of 225,964 negative sequences were used for DeepPASS-fixed training.

2. The selection of the negative training dataset may prevent the model from learning more complicated sequence contexts of PASs. As the negative sequences are sampled only from the intergenic region, they are not reflecting the actual sequence compositions in CDS or UTR regions. It would be beneficial to select negative sequences near the PASs sites or at least include some of them in the negative training dataset.

Thanks for this thoughtful suggestion. In this revised manuscript, we have now trained the DeepPASS model with negative training set containing certain amounts (from 0% to 25%) of CDS or UTR sequences as suggested. As shown in Rebuttal_Fig. 11, very minor changes were observed among AUC values when using different negative training sets with 0% to 25% CDS/UTR sequences for DeepPASS model training. Since little difference was observed among AUC values with different amounts (0% to 25%) of CDS/UTR sequences as negative training sets, we decided to keep the original data as is in the revised manuscript.

Rebuttal_Fig. 11. Only slight differences were observed with the inclusion (0-25%) of CDS/UTR sequences in negative training set for DeepPASS model training.

3. The difference between DeepPASS and DeepPASS-fixed is constructing positive training dataset. It seems that the positive sequences in DeepPASS training dataset have a broader region from the center of PASs than the positive sequences in DeepPASS-fixed. Would it be possible to expand the input sequence window (for example, from 200bp to 400bp) in DeepPASS-fixed and see whether the performance is improved or not?

Thanks for this suggestion. We have now expanded the sequence window from 200 bp to 300 and 400 bp, respectively, for both DeepPASS and DeepPAS-fixed training. For DeepPASS, the AUC values remained almost identical with the sequence length ranging from 200 to 300 and 400 bp. For DeepPAS-fixed, the AUC values slightly increased with the sequence length increasing from 200 to 300 and 400 bp, but still lower than those of DeepPASS

(Rebuttal_Fig. 12).

This result suggested that DeepPASS had better position-insensitive prediction than DeepPAS-fixed.

Rebuttal_Fig. 12. DeepPASS has better position-insensitive prediction than DeepPAS-fixed trained with longer sequence.

4. In "Comparison of DeepPASS with other models" section, do the 50,000 sequences contain the sequences presented in the training dataset of DeepPASS? Also, what is the standard error if the analysis is randomly repeated a few times?

Thanks for the questions. In the revised manuscript, stringent PASs were randomly split with a 9:1 ratio into two subgroups: training set ($n = 225,964$, 90% of stringent PAS sites) and validation set ($n = 25,107$, 10% of stringent PAS sites). The validation set is NOT used for the DeepPASS model training, and is only used for the comparison of different tools. In addition, in the revised manuscript, we randomly selected 15,000 of sequences around stringent PAS sites from the validation set ($n = 25,107$), which are NOT used for the DeepPASS training, to evaluate DeepPASS, DeepPASTA and APARENT for five times.

As indicated in Rebuttal_Fig. 13, DeepPASS shows consistently higher AUC values than DeepPASTA and APARENT, all with very low standard errors.

Rebuttal_Fig. 13. Five times of independent validation were performed to show consistent AUC values with very low standard errors for all three models. Of note, DeepPASS shows consistent higher AUC values than DeepPASTA and APARENT.

5. In step 2 "PAS evaluating with an embedded deep learning neural network", please specify the threshold of the DeepPASS prediction for calling high-confidence PASs?

We thank this reviewer for raising this thoughtful comment, and we apologize for not having clearly described the protocol in our previous submission. The threshold of the DeepPASS prediction is 0.5. Specifically, the DeepPASS model generates a predicted

probability for a given candidate PAS peak, which ranges between 0 and 1, and the candidate PAS peak is further classified into positive class (predicted probability > 0.5) or negative class (predicted probability ≤ 0.5). The DeepPASS model could achieve a sensitivity score of 0.69, a specificity score of 0.66, and an F_1 score of 0.74 when using enrichment scores for “actual” PAS analysis (Rebuttal_Fig. 6). Meanwhile, we also showed that SCAPTURE has a higher sensitivity ($\sim 71\%$ and 70%) in PASs calling than Sierra and scAPA (60% and 47%, respectively) with an independent analysis (Rebuttal_Fig. 7). These results together indicated the reliability of using SCAPTURE for PAS calling from 3' tag-based scRNA-seq.

6. Figure 1C legend does not match figure (top/middle/bottom left/middle/right

Thanks, and a new Fig. 1C is included in this revised manuscript.

7. Please specify with more details in the method section what was done during "manual subgrouping" to re-cluster the "unassigned cells".

Thanks for the suggestion. We have now added more details about how to re-cluster unassigned cells in the “method” section of our revised manuscript. Briefly, cells with different cell type assignments from DGE to DTE analyses were relabeled as distinct cell groups (“Unassigned to NKCs” and “Unassigned to DCs”). These relabeled cell groups, together with other cell types, were further investigated for marker gene expression and phylogenetic analysis. Please see our revised “methods” section.

Second round of review

Reviewer #1: Thank you for addressing my concerns.

Evaluation. Has the author satisfactorily responded to your previous review?

Reviewer #1: Yes